# Recent Advances in Anti-Metastatic Approaches of Herbal Medicines in 5 Major Cancers: From Traditional Medicine to Modern Drug Discovery

**DOI:** 10.3390/antiox10040527

**Published:** 2021-03-27

**Authors:** Jinkyung Park, Dahee Jeong, Meeryoung Song, Bonglee Kim

**Affiliations:** 1College of Korean Medicine, Kyung Hee University, Seoul 02447, Korea; pjk3607@khu.ac.kr (J.P.); jdhnsm@khu.ac.kr (D.J.); mrsong325@khu.ac.kr (M.S.); 2Department of Pathology, College of Korean Medicine, Kyung Hee University, Seoul 02447, Korea; 3Korean Medicine-Based Drug Repositioning Cancer Research Center, College of Korean Medicine, Kyung Hee University, Seoul 02447, Korea

**Keywords:** herbal medicine, cancer metastasis, lung cancer, colorectal cancer, gastric cancer, liver cancer, breast cancer, epithelial mesenchymal transition, reactive oxygen species, angiogenesis

## Abstract

Metastasis is the main cause of cancer-related death. Despite its high fatality, a comprehensive study that covers anti-metastasis of herbal medicines has not yet been conducted. The aim of this study is to investigate and assess the anti-metastatic efficacies of herbal medicines in the five major cancers, including lung, colorectal, gastric, liver, and breast cancers. We collected articles published within five years using PubMed, Google Scholar, and Web of Science with “cancer metastasis” and “herbal medicine” as keywords. Correspondingly, 16 lung cancer, 23 colorectal cancer, 10 gastric cancer, 10 liver cancer, and 18 breast cancer studies were systematically reviewed. The herbal medicines attenuated metastatic potential targeting various mechanisms such as epithelial mesenchymal transition (EMT), reactive oxygen species (ROS), and angiogenesis. Specifically, the drugs regulated metastasis related factors such as matrix metalloproteinase (MMP), serine-threonine protein kinase/extracellular regulated protein kinase (AKT/ERK), angiogenic factors, and chemokines. Overall, the present study is the first review, comprehensively investigating the anti-metastasis effect of herbal medicines on five major cancers, providing the experimental models, doses and durations, and mechanisms. Herbal medicines could be a potent candidate for anti-metastatic drugs.

## 1. Introduction

Cancer is an abnormal and uncontrolled growth of cells. There are numerous types of cancer and they have different symptoms which cause varying levels of fatality. Among them, lung cancer is the main cause of death in cancer patients (18.4% of total cancer deaths), closely followed by colorectal cancer (9.2%), gastric cancer (8.2%), liver cancer (8.2%), and breast cancer (6.6%) [1]. Moreover, lung, breast, and colorectal cancer are the respective top three cancers in terms of incidence [1].

### 1.1. Cancer Metastasis

Cancer metastasis is the process by which cancer cells spread from a primary site and progressively invade and proliferate in distant organs [2]. Metastasis is an important factor affecting tumor development and the leading cause of cancer-related deaths [3]. In over 90% of cases, metastatic expansion of cancer cells is the greatest contributor to cancer mortality [2,4]. Cancer metastasis can be induced by various pathways such as an epithelial mesenchymal transition (EMT), angiogenesis, and ROS [5,6].

#### 1.1.1. Epithelial Mesenchymal Transition

EMT allows a polarized epithelial cell to undergo multiple biochemical changes that enable it to assume a mesenchymal cell phenotype and greatly increase production of extra-cellular matrix (ECM) components [7]. In the context of neoplasia, EMT of cancer cells elevates tumor-initiating and metastatic potential, which enhances migratory capacity, invasiveness and resistance to apoptosis [8]. Tumor cell invasion is the active process of translocation of neoplastic cells across ECM barriers [9].

#### 1.1.2. Angiogenesis

Angiogenesis, the recruitment of new blood vessels, provides essential metastatic route by which tumor cells enter the circulation away from the primary tumor site [10]. ‘Angiogenic switch’, through breaking the local balance of proangiogenic and anti-angiogenic factors, is the initial step for tumor development and metastasis. Thus, inhibition of proangiogenic factors such as vascular endothelial growth factor (VEGF) and control in the physiological levels of some endogenous inhibitors comprise the strategies for anti-angiogenic agents [5,11,12,13,14].

#### 1.1.3. Reactive Oxygen Species

ROS, including hydrogen peroxide, superoxide, and hydroxyl free radicals, are mainly produced during oxygen-consuming metabolic reactions [15]. Moderate increases of ROS contribute to several pathologic conditions, among which are proliferation of malignant cancer cells and formation of metastatic colonies [16,17]. ROS activate diverse signaling pathways involving mitogen-activated protein kinase/extracellular signal-regulated protein kinases 1/2 (MAPK/ERK1/2), p38, c-Jun N-terminal kinase (JNK), and phosphoinositide-3-kinase/protein kinase B (PI3K/AKT) [18]. These, in turn, activate the nuclear factor kappa B cells (NF-κB), matrix metalloproteinases (MMP), and VEGF. Furthermore ROS can induce DNA mutation [18]. On the other hand, high-level of ROS, which can be reached by several anti-cancer treatments, suppress tumor metastasis by destroying cancer cells because of the oxidative nature of the molecules [16]. Hence, there are two ROS targeting strategies for impeding tumor angiogenesis and metastasis, to either augment tumorigenesis or lead to apoptosis. First, drugs like melatonin have a great antioxidant capacity to suppress ROS preventing activation of pro-tumorigenic signaling pathways and have synergy with some drugs enhancing its activity [19,20]. The other are drugs such as Nimbolide that induce excessive ROS generation, thereby inhibiting proliferation and metastasis via mitochondrial-mediated apoptotic cell death [21].

### 1.2. Conventional Treatment for Metastasis

As an existing treatment for metastasis, pembrolizumab is an Food and Drug Administration (FDA) approved drug for patients with metastatic non-small cell lung cancer (NSCLC) [22]. It blocks the binding of programmed cell death protein 1 (PD-1) molecule to its ligands whose interaction can inhibit anti-tumor immune response and induce metastatic progress [23]. However, pembrolizumab can induce the immune-related adverse events and may affect several organs or cause rare but serious oral mucositis [23,24]. Similarly, Nivolumab is a PD-1 inhibitor approved for the use in treatment of multiple tumor types such as melanoma, NSCLC and metastatic colorectal cancer. Nivolumab also blocks the binding of PD-1 to programmed death ligand-1 and -2 (PD-L1 and -2) which promotes T-cell proliferation and cytokine production. Still, the most common side effects of this substance in solid tumors were rash and pruritus, which occurs in more than 20% of patients [25]. On the other hand, in terms of metastatic small cell lung cancer (SCLC), platinum-based chemotherapy, and some FDA-approved immunotherapy such as atezolizumab and durvalumab consist the first-line treatment. However, there had been no FDA-approved second line therapy for patients who are resistant with platinum until 2020. Lurbinectedin, the first second-line drug approved recently in 2020, has some anti-proliferative and cytotoxic activity in multiple tumor cells and ameliorates the cells with defects in DNA mismatch repair. Nevertheless, adverse reactions such as fatigue, pneumonia, dyspnea, respiratory traction infection and musculoskeletal pain are reported at the level of grade 3 or more [26]. Considering the limitations, possibilities of resistance, and some adverse effects of those conventional treatments, a novel approach for cancer metastasis treatment is requested.

## 2. Herbal Medicines and Natural Antioxidants

Based on the recognition of these side effects, many studies have been conducted using herbal medicine for metastatic cancer. Herbal medicine has been used in clinical treatment for thousands of years in China, Japan, Korea, and other countries [27]. There are numerous researches that focused on the therapeutic effect of herbal medicine on metastatic cancer. Peng et al. proved that cancer therapy using Berberine, Curcumin, Resveratrol and other bioactive compounds extracted from medicinal herb targets microRNAs to regulate tumor growth and metastasis [12]. Zhai et al. expected the potential role of β-Elemene extracted from the herb *Curcuma Rhizoma* in curing cancer metastasis [28].

Natural antioxidants are widely distributed in medicinal plants which exhibit a wide range of biological effects, including anti-cancer, anti-inflammatory, anti-aging, and anti-atherosclerosis [29]. Spiridon et al. and Husein et al. reported that phenolic content from extracts of a number of herbal plants showed positive correlation with its antioxidant activity [30,31]. Antioxidant capacities of the herbal medicine such as Qizhu Tang, Shengmai San, *Dracocephalum heterophyllum* Benth, *Euryale ferox* Salisb. were also observed [32,33,34,35]. Free radicals pose a threat of damaging DNA, proteins, and cellular membranes, leading to onset and progress of cancer [36]. Thus, antioxidants can play a role in neutralizing abnormal cancer cell division and DNA damage [37]. This raises up the possibilities that antioxidant-rich natural products can be used in metastatic cancer treatment. Alam et al. showed that antioxidative fraction of White Mulberry induced apoptosis of cancer cell by regulating p53 and NF-κB signaling [38]. Ultimately, these lead to a firm belief that natural antioxidant, which attenuates oxidative stress, is a key component of metastatic cancer therapy and prevention.

Due to the highly metastatic conditions of lung, colorectal, gastric, liver, and breast cancer, patients with these five major cancers are exposed to high mortality. Moreover, limitations still exist in conventional medicines, such as side effects and relapse [39,40,41]. Among the previous studies, systematic reviews have been conducted on individual cancers [42], but there has not been a comprehensive study on the five major cancers with high mortality rates worldwide. In addition, the number of research regarding the anti-metastatic effect of herbal medicine is limited and the works are not organized compared to a large number of cancer cachexia studies that have been reported in a wide variety with well-structured forms [43]. Therefore, we aimed to review the anti-metastatic mechanisms of herbal medicines on five major cancers (lung, colorectal, stomach, liver, and breast).

## 3. Methods

The articles were collected using PubMed, Google Scholar, and Web of Science with “metastatic cancer” and “herbal medicine” as keywords. We included the articles showing the anti-metastatic effect of herbal medicines on the five major cancer types. The cases other than metastatic cancer and herbal medicine studies were excluded. In addition, studies using single herbal extract or mixture extract were included while, studies using single compound were excluded. Other specific exclusion criteria are presented in Table 1. The date of the references is between January 2015 and November 2020.

## 4. Results

A total of 79 studies, including 17 lung cancer, 23 colorectal cancer, 10 gastric cancer, 11 liver cancer, 18 breast cancer studies, were selected. The present study was conducted by sequentially analyzing metastatic cancers with a high mortality rate.

### 4.1. Lung Cancer

Lung cancer is the most commonly diagnosed cancer (11.6% of the total cases) and the leading cause of cancer death (18.4% of the total cancer deaths) [1]. Metastatic spread of cancer to distant organs is the reason for most cancer deaths. Lung cancer frequently metastasize to bone, brain, lung, and liver, causing a shorter survival [44]. Since chemotherapy can cause various side effects, herbal medicine is being widely used as an alternative of it, or in combination with it. Thus, in the present study, 17 studies of metastatic lung cancer treated with herbal medicine were investigated (Table 2).

Huang et al. demonstrated that *Antrodia cinnamomea* ethanol extract (ACEE) effectively inhibited angiogenesis in in vitro and in vivo. ACEE effectively suppressed the phosphorylation of signal transducer and activator of transcription 3 (STAT3) and the expression of janus kinase 2 (JAK2) by down-regulation of VEGF receptor 2 (VEGFR2) phosphorylation and suppression of downstream signaling pathways, such as phospholipase C gamma (PLCγ), focal adhesion kinase (FAK), Src, and AKT [45]. Im et al. confirmed that ethanol extract of baked *Gardeniae Fructus* (EBGF) dramatically inhibited the phorbol 12-myristate 13-acetate (PMA)-induced MMPs and urokinase-type plasminogen activator (uPA) activities in HT1080 cells via suppression of NF-κB activation. Additionally, EBGF disrupted metastasis both in in vitro and in vivo studies [46]. Wihadmadyatami et al. elucidated that *Ocimum sanctum* (OS) Linn. ethanol extract suppressed the proliferation of A549 cells and reduced the expression of integrin αvβ3, MMP-2, and MMP-9. Consequently, inhibition of migration and angiogenesis of A549 cells were observed. As a result, OS ethanol extract is suggested as a potential application of pulmonary metastasis treatment [47]. Breeta et al. reported that the *Thuja orientalis* L. extract at different concentrations has the ability to inhibit the cancer cell growth and progression by inhibiting angiogenesis. Among all the concentrations analyzed, 0.6 and 0.3 mg/mL had shown promising results with less embryo toxicity and teratogenicity [48]. Huang et al. suggested that *Viola Yedoensis* extract (VYE) regulated MMP-2 and uPA expression via the down-regulation of NF-κB pathway. Due to this process, VYE potentially suppressed the invasion, migration, and proteinase activities of A549 and Lewis lung carcinoma (LLC) cells, which consequently contributed to the inhibition of lung cancer cell metastasis [49]. Fan et al. confirmed the capability of BushenShugan Formula (BSF) to inhibit proliferation, invasion, and cancer stem cell (CSC) properties of A549 cells, decrease colony formation, and induce cell apoptosis. PI3K/AKT/NF-κB pathway may play an important role in BSF-induced inhibition of EMT process and neural-cadherin (*N*-cadherin), Vimentin, Snail, and alpha smooth muscle actin (α-SMA) expression [50]. Luo et al. indicated that Feiji Recipe inhibited the tumor progression and prolonged survival in a mouse orthotopic lung tumor model by regulating the immune response. Feiji Recipe effectively inhibited apoptosis of T cells and decreased the proportion of regulatory T cells (Tregs) by down-regulating the expression of Indoleamine-2,3-dioxygenase (IDO) [51]. Liu et al. demonstrated that Fei-Liu-Ping (FLP) ointment reduced tumor growth and metastasis in a Lewis lung xenograft mice model through the cyclooxygenase 2 (COX2) pathway. FLP suppressed the expression of COX2 in a time-dependent manner. Moreover, FLP inhibited the expression of N-cadherin, MMP-9, and vimentin [52]. Li et al. revealed that Fuzheng Kang-Ai (FZKA) may inhibit lung cancer metastasis via the STAT3/MMP-9 pathway and EMT. In addition, FZKA inhibited the growth, migration, and invasion of the lung cancer H1650, A549, and PC9 cell lines [53]. Sun et al. showed that JinFu-An (JFA) decoction administered in H1650 cells inhibited the proliferation, adhesion, migration, and metastasis of it. JFA decoction down-regulated p120ctn or its isoform 1A expression, and at the same time, up-regulated Kaiso. This might be mediated by JFA decoction targeting the lower expression of p120ctn S288 phosphorylation [54]. He et al. provided an experimental evidence that Jinfukang suppressed the differentiation and migration of lymphatic endothelial cells (LECs) in vitro, which is closely related to the tumor lymph-angiogenesis. This inhibitory effect might be via regulation of stromal cell-derived factor-1 (SDF-1)/C-X-C chemokine receptor type 4 (CXCR4) and vascular endothelial growth factor (VEGF)-C/VEGF receptor-3 (VEGFR)-3 axes [55]. Que et al. elucidated that Jingfukang can induce DNA damage in circulating tumor cell (CTC)-TJH-01 cells through the ROS-mediated ataxia telangiectasia mutated (ATM)/ATM and Rad3-related (ATR)-p53 signaling pathway. Furthermore, the proliferation and colony formation of the CTC-TJH-01 cells was arrested in the G1 phase through the p53-p21 signaling pathway [56]. Yao et al. proposed that main mechanism of JP-1-mediated anti-cancer effects is activation of p53/miR34a axis. JP-1 activated p53 and its downstream targets such as microRNA 34a (miR-34a) which leads to down-regulation of cyclin-dependent kinase 6 (CDK6), sirtuin 1 (SIRT1), c-Myc, survivin, Snail, and anexelekto (AXL) [57]. Kim et al. have clearly demonstrated that decoction MA128 inhibited the cell proliferation via p38 and ERK activation. Moreover, metastasis and tumor-induced angiogenesis of malignant tumor cells were suppressed by inhibiting the production of MMP-9 and pro-angiogenic factors in in vitro and in vivo, with no apparent side effects. These results suggest that MA128 suppresses tumor growth and present the metastatic potential of highly malignant tumor cells [58]. Xie et al. affirmed that Qingzaojiufei decoction (QD) inhibited lung tumor growth and proliferation both in in vivo and in vitro studies. Additionally, QD intensified the anti-tumor activity of cyclophosphamide (CTX). The ERK/VEGF/MMPs signaling pathways were suggested as the underlying mechanism of QD inhibiting malignant tumor cell proliferation [59]. Wang et al. reported that invasion and migration abilities of A549 cells were inhibited after Xiaoai Jiedu Recipe (XJR) treatment by reducing the expression of MMP-2 and MMP-9. Besides, XJR affected the expression of apoptosis-related proteins and decreased the expression of p-p38, p-ERK, and p-JNK. In other words, XJR promoted apoptosis and suppressed proliferation of A549 cells by blocking p38/MAPK pathway [60]. Zhao et al. showed that Xiaoji decoction (XJD) inhibited growth of lung cancer cells by increasing phosphorylation of AMP-activated protein kinase alpha (AMPKα). AMPKα-mediated inhibition of Sp1 leads to reduction in DNA methyltransferase 1 (DNMT1) protein expression. The negative feedback regulation loop of AMPKα further indicates the crucial aspect of DNMT1 in regulating the overall reactions of XJD in this process [61]. Schematic diagram of anti-metastatic mechanisms of each herbal medicine in lung cancer is elucidated in Figure 1. 

There were a total of 17 studies about metastatic lung cancer treated with herbal medicine. Overall, five of them were experimented with a single extract, and 12 were done by a mixture of extracts. Interestingly, there were two studies about a decoction Jinfukang (JFK). One of the studies proved the anti-cancer activities of JFK by directly adding oral liquid freeze-dried powder, which is a simply purified component of crude drugs in in vitro. This is more usual and traditional way of allocating natural antioxidants into the culture system [56]. On the other hand, the other study used JFK-containing serums which were prepared from JFK-dosed rats and applied to the in vitro assays. Using serum drug concentration might directly reflect the drug efficacy since crude herbal medicine exhibit complex compositions and need series of biotransformation before performing their pharmacological actions [55]. As with experiment methods, there were seven in vitro studies while the other nine used both the in vitro and in vivo. There was a single study that conducted only in vivo study. In in vitro studies, the most commonly used lung cancer cell line was A549, which was used eight times. Next was Lewis lung carcinomas cell, which was used four times. HT1080, H1650, and PC9 cell were used twice, and human umbilical vein endothelial cell (HUVEC), LEC, and CTC-TJH-01 cell were used once. In in vivo studies, C57BL/6 mice were used the most, for six times, followed by Sprague Dawley (SD) rats and zebra fish, both used twice.

There are several weak points in these studies. First of all, the level of experimental concentration of in vitro studies exceeding 60 µg is one of the shortcomings of some papers. In addition, the level of concentration was not clarified evidently in the experiments conducted with ACEE and Feiji recipe [45,51]. Furthermore, ACEE contains mainly lipids (91.4%) and only a relatively small amount of carbohydrates (2.5%). While most of previous ACEE studies have focused on the effects of carbohydrates, future studies are needed to explore the effects of lipids, which are the main ingredients of ACEE [45]. The inhibition of angiogenesis observed in zebrafish after treatment with *Thuja Orientalis* L. extract may be via inhibiting the molecular expressions of angiogenic proteins like VEGF and angiopoietin 4 (ANG4). However, further molecular studies are necessary to explore the molecular mechanism of those specific action [48]. BSF could effectively inhibit the proliferation and CSC properties of A549 cells in time- and dose-dependent manners. However, the inhibition of EMT through PI3K/AKT/NF-κB pathway worked better with cisplatin (DDP) than using BSF alone [50]. Feiji Recipe did not significantly inhibit the proliferation of 2LL, 2LL-EGFP, and 2LL-EGFP-IDO cells. Further experiments are required to clarify the specific effects and mechanisms of inhibiting tumor proliferation and stabilizing tumor focus [51]. Generally, future studies should be conducted to confirm the anti-lung cancer activity of herbal medicine in clinical level.

### 4.2. Colorectal Cancer

Colorectal cancer (CRC) is ranked within the top five in terms of high mortality and incidence due to metastasis. According to Global Cancer Observatory: CANCER TODAY (GLOBOCAN) 2018 database, colorectal cancer is the second leading cause of cancer-related deaths worldwide (881,000 deaths, 9.2% of the total) [1]. The major cause of mortality in colorectal cancer patients is distant metastasis to other organs, such as the liver and lung [62]. These days, despite advances in chemotherapy, the incidence and mortality in patients with metastatic colorectal cancer (mCRC) is still high [1]. In the present study, 23 studies related with mCRC to systematically evaluate the efficacy of herbal medicine were investigated (Table 3).

Han et al. demonstrated the anti-cancer and anti-metastatic effects of *Arctii Fructus* extract (AF) on colorectal cancer cells. AF reduced cell proliferation through apoptosis and cell cycle arrest in CT26 cells. Especially, the inhibitory effect of AF on the invasion ability of CRC cells was mediated via the AMPK signaling pathway [63]. Yue et al. demonstrated the inhibitory effects of *Scutellaria barbata* water extract (SBW) on colorectal cancer cells growth and migration in vitro, as well as the tumor growth and metastasis in colorectal tumor-bearing mice. In addition, lung metastasis inhibitory effects of SBW were reported. Most of all, this study, firstly, reported the anti-metastasis related proteins such as E-cadherin, Tspan 8, CXCR4, Src kinase by SBW [64]. Son et al. suggested that *Coix lacryma-jobi var. ma-yuen* Stapf sprout extract (CLSE) suppressed the tube formation induced by HUVECs. The underlying mechanism is the repression of the ERK1/2 and AKT pathways by CLSE under hypoxic conditions [65]. Kim et al. reported that allergen-removed *Rhus verniciflua* stokes extract (aRVS) combined with Dokhwaljihwang-tang reduced lymph nodes and the nodule in the lung previously after seven weeks. Additionally, it showed the regression of lung metastasis and very few side effects for over two years of herbal medicine treatment [66]. Zong et al. found that *Astragalus Atractylodes* mixture (AAM) could effectively inhibit migration and vasculogenic mimicry (VM) formation by suppressing ROS/HIF-1α/MMP2 pathway in HCT-116 and LoVo cancer cells under hypoxic condition. Furthermore, AAM significantly inhibited metastasis of colorectal cancer in murine lung-metastasis mice model [67]. Chen et al. demonstrated that Dahuang Zhechong Pill (DZP) inhibited the liver metastasis of CRC by suppressing CCL2 mediated M2-skewing paradigm in MC38-EGFP cells inoculated C57BL/6J mice. In addition, it was reported that DZP attenuated the accumulation of fibronectin and collagen [68]. Lee et al. reported the inhibitory effect of Danggui-Sayuk-Ga-Osuyu-Saenggang-Tang (DSGOST) on tumor necrosis factor-α (TNF-α) mediated invasion and migration of colorectal cancer HCT116 cells. DSGOST suppressed TNF-α induced nuclear translocation of Snail via inhibition of AKT/glycogen synthase kinase 3β (GSK-3β) pathways and reversed TNF-α mediated protein expression [69]. Zhu et al. reported that patients in the Chinese herbal medicine (CHM) group had a longer median survival time (40 months) than those in the non-CHM group (12 months). Moreover, the experimental results showed that aqueous extracts of 18 herbs had obvious inhibitory effects on cell proliferation and migration. Especially, they significantly decreased VEGFA, phosphorylated erythroblastic oncogene B-2 gene (p-ERBB2), p-AKT and p-VEGFR at the dose of 300 and 400 µg/mL, suggesting that anti-cancer effects were improved with increased dosages [70]. Lin et al. reported that treating both chemotherapy and Jianpi Jiedu (JPJD) can improve the mean survival time and median survival time in stage II and III CRC patients, compared with treating only chemotherapy [71]. Peng et al. found that JPJD significantly inhibited HCT116 cell viability and induced apoptosis in in vitro. It also effectively suppressed tumor cell migration, invasion, and angiogenesis by inhibiting the mammalian target of rapamycin (mTOR)/HIF-1α/VEGF signaling pathway. On the other hand, it significantly inhibited HCT116 tumor growth in athymic nude mice in vivo [72]. Liu et al. demonstrated that JPJD could inhibit transforming growth factor beta (TGF-β) induced EMT, as well as the invasion and metastasis in LoVo cell lines. Moreover, the results demonstrated that JPJD could significantly inhibit the liver and lung metastasis of orthotopic CRC tumor in BALB/c nude mice, as well as significantly prolonging the survival time of tumor-bearing mice [73]. Wang et al. showed that long-term usage of the modified Anti-cancer Decoction II Formula not only has a positive effect on the survival time of CRC patients in stage III–IV, but also helps reduce the risk of recurrence and metastasis of CRC [74]. Zhou et al. reported that Modified Si-Jun-Zi (SJZ) decoction could increase the survival rate and reduce CRC liver metastasis in GFP-HCT-116 inoculated BALB/c nude mice model by activating the innate immune system. Additionally, SJZ enhanced certain plasma cytokines such as granulocyte macrophage-colony stimulating factor (GM-CSF), increasing the number of macrophages in the spleen [75]. Kim et al. reported that Onbaekwon (OBW) exerted potent anti-metastasis effects by inhibiting expression of CXCR4, irrespective of various cancer cell types. Furthermore, suppression of CXCR4 led to down-regulation of invasion induced by the C-X-C chemokine ligand 12 (CXCL12) ligand in HCT116 colon cancer cells [76]. Lin et al. reported that Pien Tze Huang (PZH) inhibited metastasis of CRC by suppressing the TGF-β/Smad pathway, by promoting the expression of E-cadherin and suppressing the expression of N-cadherin for the first time. It also showed inhibitory effect of PZH on liver metastasis [77]. Shen et al. demonstrated that PZH inhibited the migration, invasion, and metastasis of HCT-8 cells via modulating TGF-β1/ZEB/miR-200 signaling network. It indicated that PZH suppressed the activation of TGF-β1 pathway and the expression of ZEB1 and ZEB2 which leads to EMT, whereas the expression of miR-200a, miR-200b, and miR-200c was up-regulated [78]. Lin et al. also demonstrated that PZH showed its anti-metastatic effects through the suppression of VEGFC-mediated lymph-angiogenesis. Notably, there were no change in cell apoptosis in response to PZH treatment, which suggests that PZH suppresses lymph-angiogenesis mainly by the inhibition of cell proliferation, migration, and tube formation but not directly by promoting apoptosis of human lymphatic endothelial cells (HLECs) [79]. Zhang et al. reported that the overall survival (OS) of the treatment group with Quxie Capsule for 3 months tended to be longer than that of the control group, while there were no significant differences in progression-free survival. It suggested that Quxie Capsule showed a significant survival benefit and could prolong the OS of mCRC patients [80]. Wei et al. elucidated that Teng-Long-Bu-Zhong-Tang (TLBZT) inhibited the lung metastasis of RKO colorectal cancer cells. It was proved by counting the number of metastatic nodules on the lung surface in BALB/c nude mice. This results indicated that TLBZT might be associated with the inhibition of HIF-1α-lysyl oxidase (LOX) and integrin αVβ3-FAK signal transduction [81]. Li et al. reported that Weichang’an (WCA) treatment inhibited colon tumor growth and liver metastasis compared with the control group. Additionally, HCT-116 cell proliferation and viability were significantly reduced after 72 h of treatment with WCA and serum extracted from rats that received WCA. It is meaningful that WCA was effective on decreasing expression of β-catenin, MMP-7 and serum carcinoembryonic antigen (CEA) levels [82]. Zhao et al. demonstrated that Xiaotan Tongfu (XTTF) decoction inhibited the process of hepatic metastasis in CRC by up-regulating insulin-like growth factor 1 (IGF-1)/IGF-1 receptor (IGF-1R) and down-regulating IGF binding protein 3 (IGFBP-3) secretion. In contrast, obesity influenced the secretion of IGFs, aggravating CRC. As a result, XTTF decoction can effectively inhibit hepatic metastasis and tumor growth, suggesting that XTTF decoction can serve as a potential treatment of obesity-associated CRC [83]. Pan et al. reported that Zuo Jin Wan (ZJW) treatment, acting like 5-hydroxytryptamine receptor 1D (5-HTR1D) antagonist, inhibited migration and invasion and induced apoptosis of SW403 cells. In addition, this study showed that ZJW treatment attenuated cell proliferation through the Wnt/β-catenin signaling pathway [84]. Xu et al. reported that longer duration of individualized traditional Chinese medicine decoction (more than one year) is associated with improved disease-free survival (DFS) and OS in stage II and III colorectal cancer patients [85]. Schematic diagram of anti-metastatic mechanisms of each herbal medicine in colorectal cancer is elucidated in Figure 2.

In the present study, a total of 23 studies were analyzed about anti-metastasis effects of colorectal cancer treated with herbal medicine. These were classified into four studies using single extracts and 19 studies using mixture extracts. According to analysis of single and mixture extracts, it was found that three studies used same JPJD decoction which is a traditional Chinese medicine compound from clinical experience as a target for metastatic colorectal cancer [71,72,73]. Additionally, PZH has shown the anti-metastasis effect of CRC in three studies repeatedly [77,78,79]. We could classify that there are various types of experiment including eight in vitro, four in vivo, six both in vitro and in vivo, six in clinical studies. In in vitro study, HCT116 was the most commonly used cell line followed by HT29, LoVo, SW48, CT26, and SW403 cells. On the other hand, all of in vivo studies were conducted on mice inoculated with colorectal cancer cell lines, especially on BALB/c nude mice model frequently. In each study, cytotoxicity experiments were conducted properly and the drug’s effectiveness mechanisms have also been well demonstrated from various perspectives. However, some studies were conducted at extremely high concentration, thus it is necessary to evaluate the utilization of clinical application [15]. In addition, one of study did not mention the duration of the drug’s administration, making it inaccurate to determine its effectiveness [68]. Meanwhile, all of the clinical studies, which were conducted on CRC patients, showed that the administration of herbal medicine had a significant effect on prolonging survival time. However, there were only four clinical studies, and three of them lacked sample numbers and randomness. In addition, these studies showed uncertainty whether the therapeutic effects were truly the results of herbal medicine [70,71,74]. In further research, more studies are required on biological mechanism, about how herbal medicine work against cancer metastasis. Furthermore, it is necessary to confirm these effects on more clinical trials.

### 4.3. Gastric Cancer

Gastric cancer is one of the leading causes of cancer-related deaths worldwide (8.2% of the total cancer deaths) [1], and mortality remains high, especially in East Asia. The treatment of gastric cancer patients is mainly radical surgical resection and chemoradiotherapy, while patients with metastatic tumor have great challenges to radical surgery and are prone to drug resistance [86]. In the present study, 10 studies about gastric cancer were investigated to assess the potential benefits of herbal medicine in its patients. In detail, herbal medicine’s types, experimental models, effects and mechanisms of each articles on gastric cancer are presented (Table 4).

Gao et al. observed that *Actinidia chinensis* Planch (ACP) down-regulated proliferation, migration and mesenchymal phenotypes in vitro and tumor growth in vivo. On the other hand, ACP up-regulated apoptosis and ferroptosis in vitro. Accordingly, ACP can be used as an effective treatment of gastric cancer [87]. Li et al. explained that salvianolic acid B (Sal B) and 15,16-dihydrotanshinone I (DHT I), each component featured by phenolics and diterpenoids from Danshen, predominantly suppressed myeloperoxidase (MPO) and NADPH oxidase (NOX) activity, respectively. MPO and NOX are the most abundant proteins and major source of ROS in neutrophils that mediate the PMA-induced formation of neutrophil extracellular traps (NET). Altogether, Danshen might be applied as an effective antioxidative agent in preventing early phase of NETs [88]. Liu et al. demonstrated that Babao Dan (BBD) inhibited migration and invasion via inhibiting TGF-β–induced EMT and inactivating the TGF-β/Smad signaling pathway in gastric cancer cell. These provide proper evidence of anti-metastasis effect to support the clinical application of BBD [89]. Nagata et al. elucidated that Daikenchuto (DKT) had anti-tumor effect against esophageal, breast, stomach, and colorectal cancer cell lines, inducing apoptosis in these cells. Especially in gastric cancer, oral administration of DKT had a tendency of reducing the growth and significantly reduced the peritoneal dissemination from gastric cancer. Additionally, they revealed that DKT exhibited a higher anti-tumor effect than other Kampo decoctions [90]. Zhu et al. proved that the suppressive effect of Jianpi Bushen (JPBS) on lung metastasis would be attributed to the reduction in protein expressions of Rac1, cell division control protein 42 homolog (Cdc42), SDF-1, and fibronectin (FN) in pre-metastatic lungs of mice model. Gene expressions of SDF-1 and FN were not significantly modulated by JPBS treatment. The discrepancies might be attributed to other levels of regulation on SDF-1 and FN expressions. Given that lung SDF-1 and FN proteins are key players in pre-metastatic niche formation, JPBS manifests a therapeutic role by preventing distinct metastasis among patients with gastric cancer [91]. Yuan et al. confirmed that modified Jian-pi-yang-zheng (mJPYZ) decoction could effectively prevent the occurrence of gastric cancer via suppressing PI3Kγ in macrophages from the result of in vitro and in vivo experiments [92]. Li et al. illustrated that Jinlong Capsule (JLC) down-regulated the expression of Bcl-2, survivin and up-regulated the expression of Bax, and executive caspase-3. Moreover, JLC significantly inhibited the proliferation of MGC-803 and BGC-823 cells in vitro by blocking the activity of cell cycle. Besides, JLC attenuated tumor growth and regulated apoptosis signaling pathways in vivo. Thus, JLC could inhibit gastric cancer growth and had an obvious pro-apoptosis effect in in vivo and in vitro [93]. Xu et al. found that Xiao Tan He Wei Decoction (XTHW) could suppress the proliferation of precancerous lesions of gastric carcinoma (PLGC) cells and block cell cycle at G0/G1 phase. Besides, XTHW induced apoptosis in in vitro and in vivo. All of these were caused by XTHW disrupting the activation of NF-κB [94]. Shi et al. demonstrated that Xiaotan Sanjie (XTSJ) decoction reduced interleukin 8 (IL-8) induced VEGFA and VEGFR1 protein expressions, as well as IL-8 induced VEGFR1 and VEGFR2 mRNA expression. This indicates that XTSJ may be a potential therapeutic candidate for the treatment of angiogenesis in gastric cancer via IL-8 linked regulation of the VEGF pathway [95]. Yu et al. demonstrated that Zi Yin Hua Tan (ZYHT) inhibited cell proliferation and promoted apoptosis in gastric cancer. Especially, they found that ZYHT may exert anti-cancer effect mainly through PI3K/AKT signaling pathway related with potential mechanism for proliferation and apoptosis. These results suggest that ZYHT can be used as a method for the treatment of developed gastric cancer [96]. Schematic diagram of anti-metastatic mechanisms of each herbal medicine in gastric cancer is elucidated in Figure 3.

There were a total of 10 studies about gastric cancer treated with herbal medicine. Only two of them were experimented by a single extract and eight were done by mixture extracts. As with experiment method, there were two in vitro, one in vivo study, and the other seven conducted both the in vitro and in vivo studies. Human gastric cancer cell line HGC-27 and BALB/c athymic nude mice were both used three times, which was most frequent. Cell lines such as BGC-823, MGC-823, and mice of strain 615 were used twice. There are a few weaknesses in the studies of metastatic gastric cancer treated with herbal medicine. First, the level of concentration of herbal medicine conducted in vitro exceeded 60 µg. Furthermore, the study of *Salvia miltiorrhiza* focused more on the effects of the compounds of it such as Sal B and DHT 1 than on the natural antioxidant itself [88]. In future, clinical trials are required to confirm the anti-gastric cancer activity of herbal medicine in humans.

### 4.4. Liver Cancer

Liver cancer takes up one of the most frequent causes of cancer-related mortality [97]. Especially, hepatocellular carcinoma (HCC) is the most common type (75–85%) of liver cancer [1]. However, it still remains poorly diagnosed and shows high recurrence rate, which leads to majority of HCC patients surviving less than 12 months. In the present study, 10 studies were investigated about liver cancer metastasis dealt with herbal medicine (Table 5).

Fang et al. revealed that *Actinidia chinensis* Planch root (acRoots) inhibited malignant biological behavior of HCC by attenuating DLX2. AcRoots showed more significant anti-cancer effects on stepwise metastatic HCC cell lines related with advanced stage and poor prognosis. Furthermore, acRoots effectively suppressed tumor growth, lung/intrahepatic metastasis, ascites and weight loss in in vivo by deactivating the EMT process [98]. Sunwoo et al. elucidated the anti-cancer effect of *Oldenlandia diffusa* (OD) by inducing apoptosis through caspase-3 pathway, as well as suppressing migration related gene expression in in vitro. Moreover, OD effectively enhanced the survival rates and some hepatic functions in in vivo, measured by aspartate transaminase (AST), alanine aminotransferase (ALT) and alkaline phosphatase (ALP) levels [99]. Al-Dabbagh et al. demonstrated that *Rhazya stricta* (RS) possesses consistent antioxidant, antiproliferative, and anti-metastatic activities. This consistency between antioxidant and anti-metastatic properties supports the application of natural antioxidants as a metastatic cancer therapy. Furthermore RS inhibited the colony and tube formation of HepG2 cells and exhibited significant arrest of cells at G1/S and G2/M phase by downregulating Cdc2 and its cyclin partners [100]. Wang et al. showed that *Salvia chinensis* Benth (SCB), whose main compound is protocatechualdehyde, had anti-metastatic potentials by regulating Wnt/β-catenin pathway and its transcription activity. Reduced expression of β-catenin inhibited the lung metastasis and cell cycle progression at G0/G1 stage [101]. Min et al. demonstrated the anti-angiogenic effect of Buyang huanwu decoction (BYHWD) in highly metastatic HCC xenograft mice model. Even though it had no significant inhibitory effect in tumor growth, BYHWD effectively normalized the tumor microenvironment and vasculature by modulating VEGF, regulator of G protein signaling 5 (RGS-5) and HIF-1α [102]. Han et al. showed that combination of *curcuma zedoary* and kelp (CZK), consisting of curcumenol and laminarin, suppressed the hepatic tumor growth and several metastasis-related protein expressions such as MMPs, VEGF, p-AKT, and p-ERK1/2 in H22-bearing mice model. Furthermore, this combination down-regulated cystathionine beta synthase [103]. Chen et al. reported a case of HCC patient showing that certain herbal formulation suppressed recurrent hepatic carcinoma and omentum metastasis. Furthermore, AFP level showed significant decrease after the application of herbal formulation [104]. Chen et al. revealed that Jie-du granule prolonged the median survival time of tracked down 177 Barcelona clinic liver cancer-stage C (BCLC-C) patients. Jie-du granule group showed significantly longer survival time compared to best supportive treatment group. This suggests the possibility that traditional Chinese medicine (TCM) exclusive usage can be an effective approach for advanced liver cancer patients [105]. Lin et al. demonstrated the anti-invasive mechanisms of Sini-san (SNS), which reversed HBx-induced vascular invasions and phosphorylation of metastasis-related factors such as MMP-9 through activator protein-1 (AP-1) and NF-κB binding [106]. Zhang et al. investigated that SongYouYin (SYY) could suppress the tumor growth and metastasis via enhancing immune functions. Immuno-elevating effect of SYY could reverse secreting TGF-β1 and recruiting Tregs or differentiating CD4 process, consequently suppressing tumor growth and metastasis. In addition, the synergy effect of SYY with swimming exercise suggests the combination of TCM and exercise as immunological therapy [107]. Hu et al. demonstrated that Yanggan Jiedu Sanjie (YGJDSJ) formulation reversed TGF-β1-induced EMT and Smad3 phosphorylation. As a result, this formulation inhibited HCC cell adhesion, migration and invasion, which were typical sequence of EMT mechanism [108]. Schematic diagram of anti-metastatic mechanisms of each herbal medicine in liver is elucidated in Figure 4.

In the present study, we analyzed 11 studies about liver cancer metastasis treated with herbal medicine. There were four single extracts and seven mixture extracts which have anti-metastatic effects in liver cancer. According to the classification of mixture extracts, these include specific herbal combinations designed for research and some of decoctions which are generally used in practical sites. Interestingly, all of herb extracts were not overlapped. Every single study treated its special herbal medicine. Besides, with regard to metastasis, three studies have been conducted to inhibit the transfer to the certain organs such as lung and omentum [98,104,107]. However, there is a limit to grasping tendency due to the small number of studies. As can be estimated from the number of above studies, herbal medicine focusing on liver cancer metastasis generally show lack of quantity despite of its mortality among major cancers. We believe the reason why there are only 11 studies of liver cancer may be one of the followings: First, the patient dies before the metastasis stage due to rapid progress of liver cancer. Or it is more likely that the cancer is developed from other parts of body than it did from the liver at the first hand [109]. In fact, we could easily find more studies in which cancer spread from other organs to the liver, compared to the metastasis began in liver. When it comes to experiment method, various types of experiments were designed including in vivo, in vitro, and clinical studies. In in vitro, cell lines with high metastatic potential such as HCCLM3 and MHCC97L were used most frequently [98,101,102]. Since there are limitations of applying various treatments in patients with advanced and poor-prognosis, studies using these highly metastatic cell lines are quite meaningful in terms of suggesting new treatments. On the other hand, some of the studies conducted both in vitro and in vivo experiments [98,99,101,110]. By carrying out both research on in vivo and in vitro, systemic and thorough consideration about anti-metastatic effects of herbal medicine was available compared to single-model studies. However, there were only two clinical studies conducted in liver cancer patients of advanced or metastatic stages [104,105]. Most of the patients in these stages usually have poor prognosis that any spontaneous regression or restoration is not expected. Besides, most of them have limitations in their existing anti-cancer treatment methods. One of the two research demonstrated that the life-span extension effect of herbal medicine is similar to that of existing anti-cancer treatment called sorafenib, which presents the possibility of herbal medicine as an alternative to chemotherapy in the advanced cancer stage [105]. While one of the two reported cases of 177 patients, the other reported just one case of patient treated with herbal medicine [104]. For general application in practical sites, additional clinical studies are requested. Furthermore, additional experiment is needed for clarifying the exact dose for the metastatic liver cancer patients [104].

### 4.5. Breast Cancer

Breast cancer is the most commonly diagnosed female malignancy worldwide [1]. While breast cancer shows higher survival rate compared to other four major cancers, it can be a leading cause of cancer-related deaths among women when the cancer advanced to other body sites, such as the lungs and bones [1]. Specifically, the 5 year relative survival rate of breast cancer decreased dramatically from 99 to 27% when the metastasis processed to distant sites [111]. The 18 studies were investigated related with the application of herbal extracts and mixture extracts on metastatic breast cancer (Table 6).

Choi et al. elucidated that *Alisma canaliculatum* ethanolic extract (ACEE) attenuated TNF-α-induced MDA-MB-231 breast cancer cell migration by inhibiting CXCR3 and CXCL10 expression in IκB kinase (IKK)-mediated NF-κB pathway. Since these molecules are significantly related with metastasis to bone, colon and renal cells, efficacy of ACEE plays an important role in suppressing breast cancer motility and metastasis [112]. Noh et al. identified the biological effect of *Ampelopsis japonica* (AJ) in signaling cascades that mediated the anti-metastatic process of the highly invasive MDA-MB 231 cells. In fact, AJ significantly suppressed the expression of MMPs and increased the tissue inhibitors of metalloproteinases (TIMP) level [113]. Luo et al. showed that *Camellia sinensis* (CS) suppressed tumor growth and induced cell apoptosis. Especially, CS reduced the tumor area and nodules that spread to lung and liver by suppressing MMP-2 and MMP-9 [114]. Lee et al. demonstrated that *Centipeda minima* ethanol extract (CME) significantly reduced cancer cell viability and suppressed cancer cell migration via AKT, NF-κB, and STAT3 signaling pathways [115]. Park et al. presented that *Cirsium japonicum* var. maackii extract (CJ) inhibited breast cancer cell viability, endothelial cell differentiation and tube formation, which are the first process for cancer migration and angiogenesis. Especially, Cirsimaritin, the major component of CJ, prevented angiogenesis by suppressing VEGF, p-AKT, and p-ERK in MDA-MB-231 cells [116]. Kaya et al. demonstrated that *Curcumae Radix* (CR) ethanol extract prolonged the median survival time of breast cancer model and inhibited lung metastasis, the most frequent aspect of breast cancer metastasis, through regulating the C-C chemokine receptor type 7 (CCR7)-AP-1(c-fos, c-jun)-MMP-9 pathway [117]. Yang et al. showed that ethanol extract of *Olden diffusa* (EEOD) suppressed breast cancer cell proliferation and induced apoptosis without any cytotoxic effects. This research identified MMPs and EMT pathways, especially caveolin 1 (Cav-1), as the primary targets to inhibit breast cancer metastasis [118]. Noh et al. elucidated the inhibitory effect of *Smilax china* L. (SCL) in proliferation and migration of MDA-MB-231. SCL markedly reduced mRNA levels of some molecules such as uPA, urokinase-type plasminogen activator receptor (uPAR), and TIMP, which are deeply associated with ECM degradations and migration of breast cancer cell [119]. Lai et al. showed that *Solanum nigrum* (SN) effectively suppressed viability of MCF-7 by inducing cancer cell apoptosis and cell cycle arrest which are mainly mediated by activation of ROS and caspase-3. In addition, SN reversed EMT process, which maximizes cancer cell growth, metastasis, and drug-resistance. Besides, these lead to some morphological changes in mitochondria, which are helpful for its metabolism and function after application of SN [120]. Noh et al. demonstrated anti-metastatic effects of *Rheum palmatum* L. extract (RPE) by modulating the ECM degradation-related proteins such as uPA, uPAR, and MMPs. RPE also affected NF-κB transcription factors by down-regulating IkBα, which consequently regulates the expression of uPA and MMPs. Furthermore, RPE regulated MAPK and AKT signaling pathway, leading to synthesizing and releasing of MMPs [121]. Lee et al. elucidated that *Toxicodendron vernicifluum* stokes extract (TVSE) suppressed not only the growth of solid tumors in breast itself but also the metastasis to lung in 4T1-injected mice. While decrease in Ki67 implied the inhibition of proliferation, transcriptional alternation of several genes such as MMP-2, TIMP-1, uPA, intercellular adhesion molecule 1 (ICAM-1), vascular cell adhesion molecule 1 (VCAM-1), and plasminogen activator inhibitor 1 (PAI-1) played key roles in suppressing metastasis [122]. Yue et al. invented an innovative herbal formula for reducing tumor growth and extending the life span of breast cancer model. Specifically, IL-12 level was augmented, while Tregs, forkhead box P3 (FOXP3+) in lymph node and some myeloid-derived suppressor cells (MDSC) in spleen were decreased. Moreover, this new formula showed anti-metastatic effects against liver, lung, and bone metastasis, especially inducing restoration of osteolysis and several series of immune response [123]. Zhang et al. elucidated that Gubenyiliu II exerted anti-tumor growth and anti-metastatic effects on breast cancer model by decreasing heparanase and growth factor expression, which subsequently led to suppressing ERK/AKT pathways. Meanwhile, components of Gubenyiliu II were once more classified in similar function groups so called ‘huoxuehuayu’, and ‘jiedusanjie’. Consequently, it was revealed that ‘huoxuehuayu’ and ‘jiedusanje’ TCM pharmacological function are related with attenuating tumor growth and metastasis [124]. Li et al. revealed the efficacy of Ruyiping (RP) in terms of suppressing the tumor growth and metastasis, whose main mechanism was related with reduction in EMT and MMP-9. Besides, RP induced arrest of G2 cell cycle, which are evidenced by remarkable decrease in CDK1 and cyclin B1 expression [125]. Ye et al. demonstrated that Ruyiping with *Platycodon grandiflorum* (RP+PG) inhibited the breast cancer metastasis to lung by protecting pulmonary vascular integrity and then suppressing the extravasation of fibrinogen. Consequently, this led to preventing the formation of inflammatory pre-metastatic microenvironment for lung metastasis [126]. Liu et al. reported that fermentation products of *Trametes robiniophila Murr* with *Radix Isatidis* (TIF) suppressed cancer cell proliferation and metastasis via p53 and MMP pathways. In addition, TIF induced apoptosis and arrest of G2/M cell cycle. As TIF showed enhanced efficacy compared to individual use, this suggests synergetic effects between herbs [127]. Ma et al. demonstrated that Wensheng Zhuanggu (WSZG) suppressed the motility and invasion in breast cancer cells by down-regulating TGF-β1/Smads signaling pathway. WSZG reversed the effects of bone marrow-derived mesenchymal stem cells (BMSC) that enhanced the bone-invasive and metastatic tendency of breast cancer cells by inducing EMT, implying the potential application as an anti-bone metastatic agent [128]. Wang et al. validated the critical role of CXCL1 in mediating anti-metastatic activities of Xiaopi formula such as inhibiting stem cells subpopulation, and suppressing metastatic micro-environment formation. While Xiaopi formula efficiently prevented breast cancer progress in mice, it showed little inhibitory effects on breast cancer cells. This superior efficacy in in vivo might be attributed to bio-systemic regulation and interaction, especially for tumor micro-environment [129]. Schematic diagram of anti-metastatic mechanisms of each herbal medicine in breast cancer is elucidated in Figure 5.

In the present study, we analyzed 18 breast cancer studies treated with herbal medicine. There were 11 single extracts and 7 mixture extracts. Single extracts are extracted from plant products with ethanol or water. On the other hand, the classification of mixture extracts includes two specific herbal combinations designed for research and five of decoctions generally used in practical sites. According to analysis of the specific source of each treatment, most of studies treated different herbal medicine. Only the decoction called RP was studied twice, which has been used as an effective traditional herbal formula for preventing post-operative recurrence and metastasis of breast cancer [125,126]. With regard to metastasis to the certain organs, there were six lung, two bone, and two liver metastasis cases. This corresponds with previous contents that breast cancer is likely to spread to the lung or bone [1].

In terms of experiment method, both in vitro and in vivo studies were conducted. Among in vitro studies, the most frequently used cell line was MDA-MB231 followed by MCF7 and 4T1. Especially, MDA-MB-231 and MDA-MB-468 are triple-negative breast cancer cell lines with high metastatic potential, which means a number of studies focus on breast cancer with poor prognosis. However, some in vitro studies did not mention exact dosage or treated quite high dosage of herbal medicine [124,125]. Thus, further research about confining concentration are required. On the other hand, in terms of in vivo studies, most of them used mice models inoculated with breast cancer cell lines. However, one of the studies used zebrafish, which have advantage in showing quick effects like cell experiments [118]. In particular, two in vivo studies validate ingredient-target network, which is recently emerging as a powerful tool for identifying the targets of therapy and understanding the complex action mechanisms of TCM formulas [118,129]. Since most of herbal medicine are used together as a combination and the interactions between them have quite important meaning in TCM, these two studies suggest a new research method for herb products [118,129]. Furthermore, one of the studies showed that fermentation of natural substances decreased the cytotoxicity and increased the anti-metastatic effects [127]. This suggests the possibility of obtaining new efficacy and minimizing side effects through fermentation of various natural products. One of the studies compared its efficacy to combined usage with a chemical medicine called zoledronate. However, all the anti-metastatic effects that were mentioned above and additional anti-osteolytic effect barely showed better effects on combined usage than on CS alone. Therefore, it raises the need for experiment design for herbal medicine alone [114]. Meanwhile, there were some studies conducting experiments both on in vivo and in vitro. Since the results showed the difference between cell models and animal models, it is necessary to conduct further research and discussion about how the difference occurred [115,129]. In other words, some studies on bio-systemic mechanisms and interactions of herbal medicine are demanded. By the way, there was a lack of clinical research regardless of numerous cell and animal studies, which practically limits the clinical utilization of herbal medicine. 

## 5. Discussion

This research is the first systematic study to comprehensively and intensively identify anti-metastatic effects of herbal medicine on the five major cancers (lung, colorectal, stomach, liver, and breast cancer). We analyzed the correlation between each cancer and herbal medicine, and the mechanism of each herbal medicine. Therefore, the research was conducted to develop the existing herbal medicine studies that were biased only in cancer cachexia, and establish a systemic database on anti-metastatic effects of herbal medicine focusing on five major cancers.

This study was conducted based on the research data between January 2015 and November 2020. Of these 79 studies, 26 used a single extract, and the remaining 53 used mixture extracts as a form of herbal medicine. Interestingly, there were more studies using mixture extracts than single extract due to the effects of herbal medicine prescriptions, which showed synergy through the combination of various medicinal herbs. In contrast, there were more studies of single extract in breast cancer. The components of every mixture extracts are arranged in Table 7.

The main efficacy associated with cancer metastasis of herbal medicine was: (1) suppression of proliferation, viability, and tumor growth; (2) inhibition of migration, invasion, and metastasis, which was shown in almost all research. Followed by (3) induction of apoptosis; (4) inhibition of angiogenesis; (5) arrest of cell cycle. Especially, AF on colorectal cancer, JLC on gastric cancer and SN, RP, TIF on breast cancer showed the efficacy of arrest of cell cycle in the G2/M stage [63,93,120,125,127]. On the other hand, Jingfukang on lung cancer, ZJW on colorectal cancer, XTHW, ZYHT on gastric cancer and SCB on liver cancer presented the efficacy of arrest of cell cycle in (G0)/G1 stage [56,84,94,96,101]. In addition, all clinical studies have shown a significant increase in survival time in the group of metastatic cancer patients administered with herbal medicine. Interestingly, we also found a correlation between each cancer and metastasis to certain organs. It was shown that five major cancers are likely to spread to the lung mostly. The study of the anti-metastasis effects of herbal medicine on the transition from specific cancer to lung showed up in all cancers. There are four studies which demonstrated anti-lung metastasis effects in colorectal cancer, one in gastric cancer, three in liver cancer, six in breast cancer. After lung metastasis, liver cancer metastasis was followed which have been spread from primary colorectal and breast cancer shown in six and two papers, respectively. Exceptionally in breast cancer, there are two studies related to bone metastasis Schematic diagram of herbal medicine used for mutual organs metastasis is shown in Figure 6.

In terms of mechanisms, a wide variety of factors were used for each herbal medicine in all five major cancers. The main mechanisms common to the five major cancers were EMT, MMPs, ROS-related process and signaling pathway. In detail, first, herbal medicines suppress metastasis by impeding the EMT-related cascades, promoting the expression of E-cadherin and suppressing the expression of N-cadherin, Snail, and Smads. Second, herbal medicines inhibit metastasis by suppressing MMPs-related cascades and signaling transduction such as AKT/ERK, VEGF, TNF-α, and TGF-β/Smad pathways. Lastly, antioxidant herbal medicines take advantage of the ROS-induced cascades showing anti-tumor effects. Especially in colorectal cancer, studies on hypoxia-related factors HIF-1α are also notable. Following that, studies on anti-metastasis effect through regulating immunity have also been conducted. To summarize, diverse factors involved in anti-metastatic mechanisms of herbal medicine can be organized in a comprehensive view as Figure 7.

A total of 79 of five major cancers studies, various types of experiments were designed including in vivo, in vitro, and clinical studies. Overall, there were 28 studies in in vitro, 13 in in vivo, 32 both in in vitro and in vivo, and 8 in clinical studies. Most of the in vivo experiments were conducted with human cancer cell lines inoculated mice model. On the other hand, two experiments using zebrafish which have advantage in showing quick effects were also found [89,118]. Especially, in breast cancer studies, triple-negative breast cancer cell lines are used that means a number of studies showed a therapeutic effect on severe metastatic cancer prognosis. Similarly, in liver cancer studies, high metastatic cell lines such as HCCLM3 and MHCC97L were used most frequently, and one of study even showed more significant anti-cancer effects compared to general HCC cell lines [98]. In addition, we found that there are only a few articles on clinical studies, which is only eight in entire articles. In particular, there are extremely wide variety of herbal medicines and each herbal medicine is lack of in-depth experiments. Hence, sometimes sufficient support is not provided to demonstrate the real potential to apply into actual clinical trials. This means that more experiments are needed on clinical utilization of herbal medicine in further studies.

Especially in colorectal cancer, several studies significantly identified the anti-metastatic effect with overlapped decoctions; three using JPJD decoction and three using PZH stressed a valid efficacy for colorectal metastatic cancer. On the other hand, all the studies in liver cancer did not overlap each other. Additionally, they showed that metastasis from other organs to the liver are more common than those began from the liver [109]. In addition, *Oldenlandia diffusa* showed anti-metastasis effect in both liver and breast cancer [99]. The study of DKT which focused on gastric cancer mostly showed anti-tumor efficacy of esophageal, breast, and colorectal cancer at the same time [90]. The comprehensive analysis, this suggests the potential application of herbal medicine to other cancer types whose anti-metastatic effects were previously validated from other specific cancer type. 

Previously, Wang et al. and Yang et al. reviewed the effects of herbal medicine as adjuvant treatment in each advanced non-small cell lung cancer and triple-negative breast cancer [130,131]. In addition, Shao et al. summarized the articles about liver cancer limited metastasis [132]. However, they showed only the effects on single cancer type or one specific way of metastasis, making it difficult to determine whether the drugs were effective other cancer species. In addition, cancer was prone to metastasize to various parts of body, it is important to understand which organ cancer spreads to. However, because it was limited to only one type of cancer and could not identify the organic relationships between various organs, it is not possible to comprehensively observe how the drug and effects are expressed depending on where the metastasis was located. In contrast, the present study has the strength in integratively analyzing five major cancers of high mortality rates. Hence, looking at the transition to various organs organically, present study makes it possible to analyze the association of each organ or the metastatic cancer tendency between five major cancers. Besides, while analyzing five types of cancer at the same time, it was possible to grasp a recent trend of herbal medicine related to metastasis. That is, while there had been many herbal extracts studies related with certain cancer species like colorectal and breast cancer, some cancer species showed quantitative deficiency of herbal medicine research. Herbal extracts that could be applied to multiple cancer species were also presented via present analysis. It is meaningful that our study provided the insight in which herbal medicine are used for cancer metastasis in common and compared and evaluated the association among the five major cancers as a whole. In addition, our research has an advantage in that the mechanisms and effects of herbal medicine are visually represented through tables and figures, compared to the other review research that did not provide an organically connected figure about mechanisms [106,133].

Furthermore, most of existing herbal medicine studies have focused on the anti-cancer effect with a single “compound” rather than a mixture form. However, due to the intra-tumor heterogeneity of cancer cells, resistance to specific-targeted compounds arises from signaling redundancies and genotype selection [134]. Besides, tumor microenvironments, which commonly appear at the onset of metastasis, modulate the efficacy of cytotoxic agents and targeted therapies [135]. Therefore, inspiration for multi-compound and multi-target anti-metastasis research is requested. Unlike single-targeted therapy, natural products, as a mixture of multiple compounds, have the potential to interact with multiple targets simultaneously [136]. Synergistic interactions within a natural compounds mixture also enhance the search for potential molecular targets in cancer cells [137]. This present study shows the efficacy of the multi-compounds and multi-target mechanisms-centered herbal. Hence, this study suggests a strategy for overcoming the limitations and some resistance of the drug application due to the single-compound drug and its limited pathways.

However, there are limitations in the present study. First of all, the research was limited to recent five years only. Additionally, experiments conducted with a specific compound of herbal medicine were excluded. In addition, more extensive research selection is demanded by using more diverse search engines other than PubMed, Google Scholar, and Web of Science. It is also necessary to supplement for studies on more diverse cancer types and metastasis areas that we could face in actual clinical sites, not only limited to five major cancers. Therefore, further research is needed to cover the studies that were excluded in our research such as compounds of herbal medicine, papers prior to 2015 and different types of cancer other than five majority cancers. Complementary studies will provide a lot more comprehensive and useful data of anti-metastasis effects of herbal medicine.

## 6. Conclusions

In conclusion, this study provides an integrated view of herbal medicine by systematically analyzing the mechanism and examining clinical practices of herbal medicine on the five major cancers in total 79 studies. The herbal medicines attenuated metastatic potential of various cancer cells targeted various mechanisms such as EMT and ROS. Specifically, the drugs regulated metastasis related protein/RNA expression such as MMP, AKT/ERK and angiogenic factors and chemokines. Due to high mortality and prevalence rates, five major cancers highly demand for conventional treatment and herbal medicine as a breakthrough. Thus, our study will be a significant database of providing a useful clinical database and the connection of effective treatment as a potent anti-cancer agent. With this database, herbal medicine could be a complementary therapy that can overcome the limitations of current anticancer drugs. In addition, this study supports the scientific evidence for novel drug development, as it revealed experimentally proven mechanisms for its efficacy. Therefore, it is expected that herbal medicines will be fully utilized in actual clinical treatment through comprehensive consideration of these mechanisms and further clinical trials in the future.

## Figures and Tables

**Figure 1 antioxidants-10-00527-f001:**
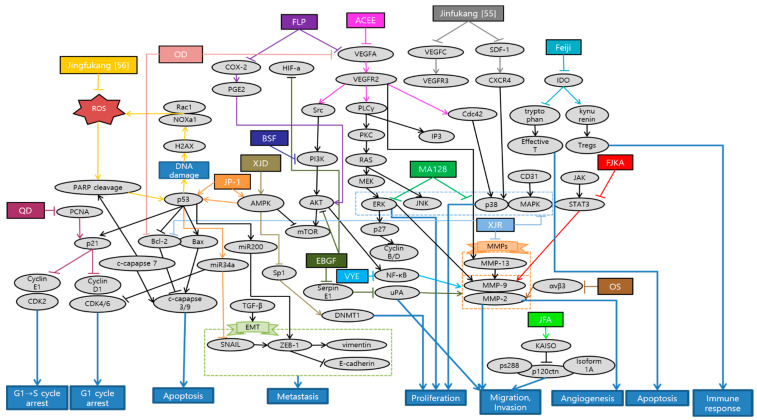
Schematic diagram of anti-metastatic mechanisms of herbal medicine in lung cancer: Herbal medicine can suppress angiogenesis, metastasis and proliferation by inactivating sequential cascades below such as epithelial mesenchymal transition (EMT) and matrix metallopeptidases (MMPs). Vascular endothelial growth factor A (VEGFA) and VEGF receptor 2 (VEGFR2) signaling activate the downstream pathways such as phospholipase C gamma (PLCγ), focal adhesion kinase (FAK), Src and protein kinase B (AKT). Cyclooxygenase 2 (COX2) and prostaglandin E2 (PGE-2) can also activate AKT, which generally begins from phosphoinositide-3-kinase (PI3K). This activates nuclear factor kappa B cells (NF-κB) affecting MMPs and EMT processes, which are the basic steps for metastasis. In the process of EMT, Snail activates zinc finger E-box-binding homeobox-1 (ZEB-1), vimentin, N-cadherin, while down-regulating epithelial cadherin (E-cadherin). On the other hand, ERK/p38/MAPK pathway which is activated from various factors such as C-X-C chemokine receptor type 4 (CXCR4), cluster of differentiation 31 (CD31), electron transport chain (ETC), also induces MMPs. Furthermore, natural antioxidants can affect reactive oxygen species (ROS)-induced cascades, promoting apoptosis, cell cycle arrest and some metastatic changes. As a reaction to DNA damage from oxidative stress, poly (ADP-ribose) polymerase 1 (PARP1) is cleaved and regulates the stability of p53. DNA damage by ROS also mediates ataxia telangiectasia mutated (ATM)/ATM and Rad3-related (ATR)-p53 signaling pathway. P53 affects B-cell lymphoma (Bcl) and Bcl-2 associated X protein (Bax) which leads to caspase 3/9 cleavage for apoptosis. Additionally, p53 activates p21 expression which is highly related to cell cycle arrest, especially G1 phase. P53/miR34a axis is proposed to induce cell cycle arrest and suppress some of the EMT process. Herbal medicine can also stimulate T cell apoptosis and suppress Tregs immune function by inhibiting indoleamine-2,3-dioxygenase (IDO) expression. Abbreviation: ACEE, *Antrodia cinnamomea* ethanol extract; EBGF, ethanol extract of baked *Gardeniae Fructus;* OS, *Ocimum sanctum*; VYE, *Viola Yedoensis* extract; BSF, BushenShugan Formula; FLP, Fei-Liu-Ping; FZKA, Fuzheng Kang-Ai; JFA, Jinfu’an; QD, Qingzaojiufei decoction; XJR, Xiaoai Jiedu Recipe; XJD, Xiaoji decoction.

**Figure 2 antioxidants-10-00527-f002:**
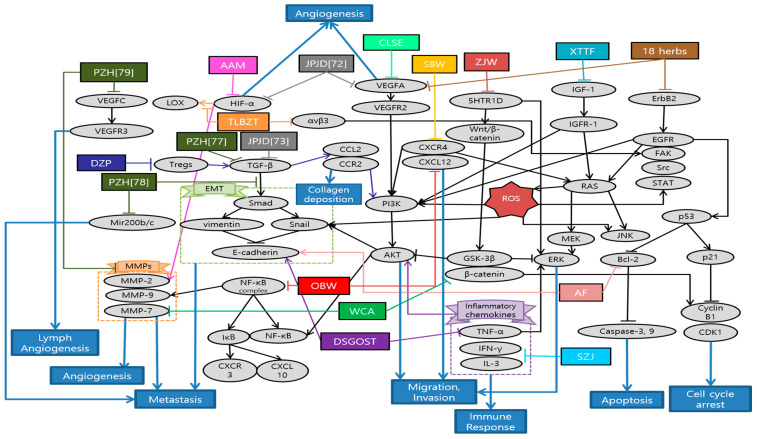
Schematic diagram of anti-metastatic mechanisms of herbal medicine in colorectal cancer: Herbal medicines suppress angiogenesis, migration and invasion via inactivation of diverse HIF-1α pathways. Tumor hypoxia results from the imbalance between the oxygen supply and demand due to the uncontrolled tumor cell proliferation. Under hypoxic condition, HIF-1α expression is accumulated as a pro-angiogenic factor, which is mainly involved in MMP-2 production. Hypoxia also activates the phosphorylation of ERK1/2 and AKT, which make cancer cell migration and invasion possible. Over-activation of mTOR, the key regulator of cancer progression, is frequently associated with activation of HIF-1α, which stimulates tumor genesis, angiogenesis, and tumor growth through VEGF. Furthermore, herbal medicine suppress metastasis by impeding EMT-related cascades. EMT refers to the development of mesenchymal cells through a series of changes in epithelial cells, playing an important role in the cancer cells growth via regulation of immunosuppressive tumor microenvironment. TGF-β activates the Smad pathway, which leads to other EMT-related protein expressions. In the same way, TGF-β1 modulates ZEB leading to EMT and miR-200 expression. TNF-α also induces nuclear translocation of Snail via AKT/GSK-3β pathway. Meanwhile, herbal medicine can serve similarly with antagonists of 5-HTR1D suppressing Wnt/β-catenin signal transduction via β-catenin degradation. 5-HTR1D influences the Wnt/β-catenin signaling at the upper stream, leading to stabilization of β-catenin in cells. MMP-7 is transcribed in response to Wnt-β-catenin signaling which assists the degradation of extra-cellular matrix (ECM), a critical event in tumor invasion and metastasis. In terms of immune regulation, herbal medicine can enhance certain plasma cytokines such as GM-CSF, increasing the number of macrophages in the spleen which contribute to the immune response and the prolongation of survival time. Abbreviation: AF, *Arctii Fructus* extract; SBW, *Scutellaria barbata* water extract; CLSE, *Coix lacryma-jobi var. ma-yuen* Stapf sprout extract; AAM, *Astragalus Atractylodes* mixture; DZP, Dahuang Zhechong Pill; DSGOST, Danggui-Sayuk-Ga-Osuyu- Senggang-Tang; 18 herbs, Formula of the aqueous extracts of 18 herbs; JPJD, Janpi Jiedu; SJZ, Modified Si-Jun-Zi Decoction; OBW, Onbaekwon; PZH, Pien Tze Huang; TLBZT, Teng-Long-Bu-Zhong-Tang; WCA, Weichang’an; XTTF, Xiaotan Tongfu; ZJW, Zuo Jin Wan.

**Figure 3 antioxidants-10-00527-f003:**
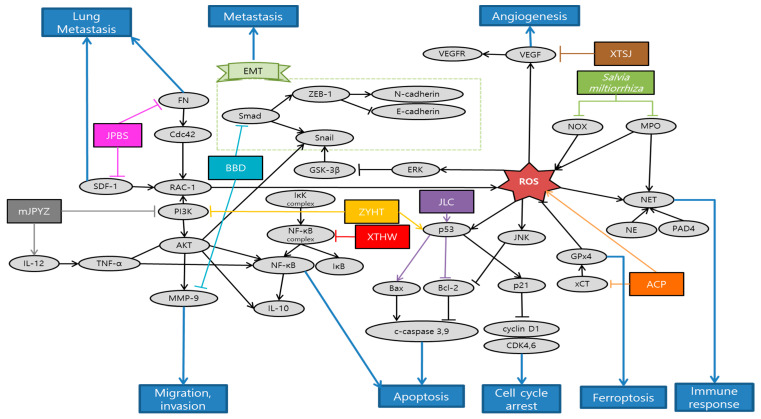
Schematic diagram of anti-metastatic mechanisms of herbal medicine in gastric cancer: When it comes to ROS, it has double-edged aspects in anti-tumor efficacy and mechanisms. Therefore, there are also two opposite ways that natural antioxidants utilize ROS as part of anti-metastatic process. First of all, ROS is predominantly generated by NOX as an early response to phagocytosis-mediated microbicidal activity of neutrophils. There exists a complicated enzymatic cascade of NOX, protein-arginine deiminase 4 (PAD4), neutrophil elastase (NE) and MPO which plays a crucial role in triggering sequential generation of neutrophil extracellular traps formation (NETosis) for circulating tumor cells and metastasis. Natural antioxidants might be applied as an effective antitumor agent in preventing early phase of NETs as a result of ROS. In contrast, herbal medicine can promote ferroptosis by maximizing ROS. Abbreviated expressions of glutathione peroxidase 4 (GPx4) and cystine/glutamate antiporter (xCT), inducing the ROS accumulation in human gastric cancer (HGC) cells 27, finally aggravate the ferroptosis of cancer cells. Furthermore, herbal medicine can inactivate the metastasis related process. TGF-β activates the Smad signaling pathway, which shares a common EMT pathway. SDF-1 and FN proteins are key factors in pre-metastatic niche formation in the lung, which lead to Cdc42 and Rac-1 expression. Besides, herbal medicines stimulate cancer cell apoptosis related processes. For instance, PI3K/AKT/NF-κB signaling pathway is related to potential mechanisms for proliferation and apoptosis. Executive caspase-3, which is activated by up-regulated Bax or down-regulated Bcl-2, can also induce apoptosis. Abbreviation: ACP, *Actinidia chinensis* Planch; BBD, Babao Dan; DKT, Daikenchuto; JPBS, Jianpi Bushen; mJPYZ, modified Jian-pi-yang-zheng; JLC, Jinlong Capsule; XTHW, Xiao Tan He Wei; XTSJ, Xiaotan Sanjie; ZYHT, ZiYinHuaTan.

**Figure 4 antioxidants-10-00527-f004:**
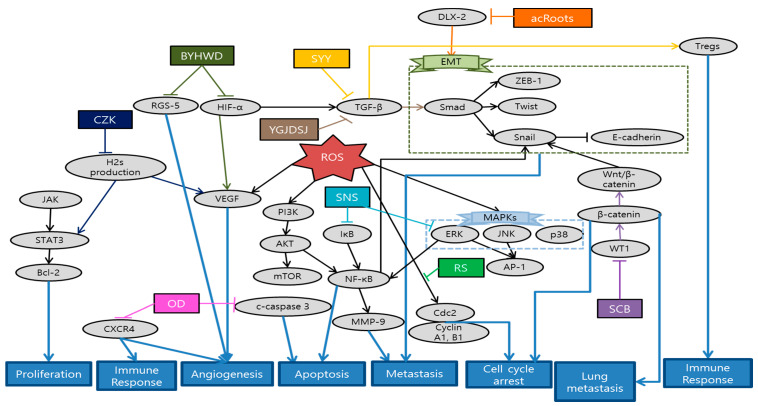
Schematic diagram of anti-metastatic mechanisms of herbal medicine in liver cancer: Herbal medicines interrupt signaling transduction and EMT cascades related with cancer cell migration and metastasis. ERK/JNK signaling pathway activates the AP-1 activity. Precedent cascades of ERK/PI3K/AKT regulate the NF-κB activity. Both AP-1 and NF-κB bind to promoter sites of MMP-9. distal-less homeobox 2 (DLX2) can activate the EMT process which generally begins with TGF-β1 by Smad3 phosphorylation. Wnt/β-catenin pathway and its transcription activity also activate Snail, composing a process of EMT. On the other hand, immuno-elevating and anti-angiogenic effects of herbal medicines can inhibit microenvironment creation for cancer metastasis. VEGF, RGS-5, and HIF-1α create some of the tumor microenvironments and vasculature. Cancer cells can induce an immuno-suppressive microenvironment that makes tumor growth and metastasis available by secreting TGF-β1 and recruiting Tregs or differentiating CD4. Abbrevation: acRoots, *Actinidia chinensis* Planch root; OD, *Oldenlandia diffusa*; RS, *Rhazya stricta*; SCB, *Salvia chinensis* Benth; BYHWD, Buyang Huanwu decoction; CZK, combination of *curcuma zedoary* and kelp; SNS, Sini-san; SYY, SongYouYin; YGJDSJ, Yanggan Jiedu Sanjie.

**Figure 5 antioxidants-10-00527-f005:**
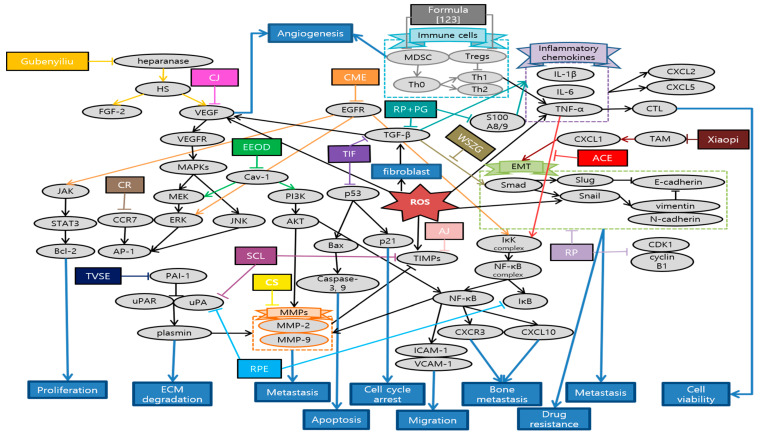
Schematic diagram of anti-metastatic mechanisms of herbal medicine in breast cancer: Herbal medicine suppress metastasis by blocking a series of EMT-related processes which starts with TGF-β1 and sequentially provoke Smad, Snail, twist, N-cadherin, and, etc. CXCL also mediates EMT which contributes to metastatic micro-environment formation. Moreover, herbal medicine can suppress the expression of MMPs, which play multiple key roles in facilitating the metastasis of tumor cells by direct or indirect pathways. In the latter, MMP-9 is stimulated by various pathways such as AKT/ERK, NF-κB, STAT3, CCR7-AP-1 pathway, mRNA modulations of uPA and TIMPs. AKT/ERK pathways are also stimulated by Cav-1 and VEGF which is the onset of angiogenesis. On the other hand, NF-κB induced by TNF-α can activate MMPs, as well as metastasis related factors such as CXCR3-CXCL10, ICAM-1 and VCAM-1. Besides, herbal medicines also stimulate cancer cell apoptosis and cell cycle arrest which are usually mediated by activation of ROS and caspase-3. Expression of CDK1/cyclin B1 and p21 stimulated from p53 were related with cell cycle arrest and apoptosis. Meanwhile, there are two opposite ways in which herbal medicine controls the immune response as anti-metastatic mechanisms. First of all, excessive inflammatory responses create a favorable environment for cancer metastasis. Recombinant S100 calcium-binding protein A8/A9 (S100A8/A9) mobilizes the expressions of pro-inflammatory cytokines including IL-1β, IL-6, and TNF-α through the production of ROS, which stimulates NF-κB in peripheral blood mononuclear cells. Furthermore, those inflammatory chemokines produce CXCL2 and CXCL5 which have strong chemotactic capacities on neutrophils. Herbal medicines suppress those excessive immune responses and prevent metastasis. In contrast, building up the overall immune system can upgrade the survival ability of patients. Natural combinations control Tregs and MDSC population, as well as increase the anti-tumor cytokine. This contributes to prolongation of life span. Abbrevation: ACE, Alisma canaliculatum ethanolic extract; AJ, Ampelopsis japonica; CS, Camellia sinensis; CME, Centipeda minima extract; CJ, Cirsium japonicum; CR, Curcumae Radix; EEOD, ethanol extract of Olden diffusa; SCL, Smilax china L.; SN, Solanum nigrum; RPE, Rheum palmatum L. extract; TVSE, Toxicodendron vernicifluum stokes extract; RP, Ruyiping; RP+PG, Ruyiping with Platycodon grandiflorum; TIF, Trametes robiniophila Murr with Radix Isatidis; WSZG, Wensheng Zhuanggu.

**Figure 6 antioxidants-10-00527-f006:**
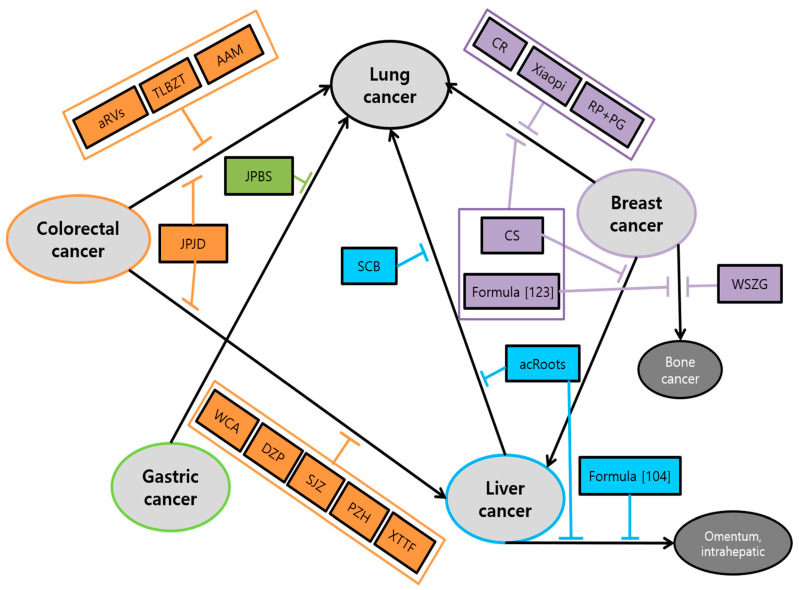
Schematic diagram of herbal medicine used for mutual organs metastasis: In addition to inhibitory actions against individual types of cancers, herbal medicine can be effectively applied to some metastatic interactions between five major cancers. Herbal medicine prevents the metastasis of the five major cancers to each other. In particular, there were a number of herbal medicines that prevent cancer from spreading to the lung. Abbrevation: aRVS, allergen-removed *Rhus verniciflua* stokes extract combined with Dokhwaljihwang-tang; TLBZT, Teng-Long-Bu-Zhong-Tang; AAM, *Astragalus Atractylodes* mixture; JPJD, Janpi Jiedu; WCA, Weichang’an; DZP, Dahuang Zhechong Pill; SJZ, Modified Si-Jun-Zi Decoction; PZH, Pien Tze Huang; XTTF, Xiaotan Tongfu; JPBS, Jianpi Bushen; SCB, *Salvia chinensis* Benth; acRoots, *Actinidia chinensis* Planch root; CR, *Curcumae Radix*; RP+PG, Ruyiping with *Platycodon grandiflorum*; CS, *Camellia sinensis*; WSZG, Wensheng Zhuanggu.

**Figure 7 antioxidants-10-00527-f007:**
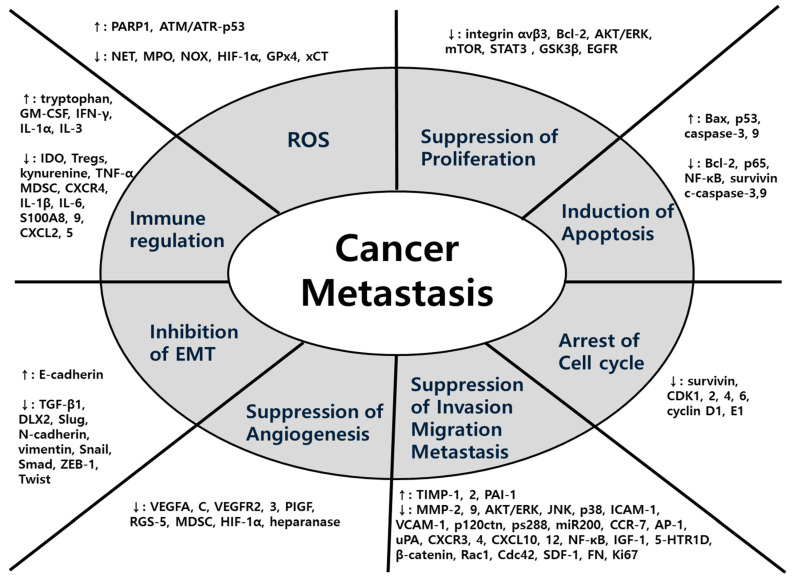
Comprehensive diagram of anti-metastatic factors regulated by herbal medicine: Herbal medicine have diverse efficacies and mechanisms against cancer metastasis. This figure shows the comprehensive view of efficacies and each specific targeting molecules.

**Table 1 antioxidants-10-00527-t001:** Exclusion criteria.

Research on Diseases Other than Cancer
Research on the effects and mechanisms of non-herbal medicines
Research on alleviation of symptoms and side effects of chemotherapy, not inhibition of metastasis
Review research
Research that used single compound
Research on early local cancer treatment and surgical treatment independent of metastasis

**Table 2 antioxidants-10-00527-t002:** Lung cancer.

Classification	Herbal Medicine	System	Experimental Model	Dose; Duration	Efficacy	Mechanism	Reference
Single Extract	*Antrodia cinnamomea* (ethanol)	In vitro	HUVEC	0.05, 0.1%; 6 h	① Suppression of proliferation and tube formation ② Inhibition of migration and angiogenesis	↓: P-VEGFR2, VEGFR2, P-PLCγ1, P-FAK, P-Src, P- AKT, P-STAT3, JAK2, CD31	[45]
		In vivo	LLC-inoculated C57BL/6 mice	0.5, 1%; 30 days			
Single Extract	Baked *Gardeniae Fructus* (ethanol)	In vitro	HT1080	250, 500 µg/mL; 24, 48 h	① Suppression of tube formation ② Inhibition of migration, invasion, metastasis and angiogenesis ③ Decrease in colony formation	↓: MMP-9, MMP-13, uPA, NF-κB, serpine E1, EGF, pentraxin, Ang-2, PDGF, IL-8, VEGF, PIGF	[46]
		In vivo	B16F10-injected C57BL/6 mice	50, 100 mg/kg; 21 days			
Single Extract	*Ocimum sanctum* Linn. (ethanol)	In vitro	A549	50, 70, 100, 200 µg/mL	① Suppression of proliferation and tube formation ② Inhibition of migration and angiogenesis	↓: integrin αvβ3, MMP-2, 9	[47]
Single Extract	Leaf of *Thuja orientalis* L. (ethanol)	In vitro	A549	0.3, 0.6, 1.2, 2.4 mg/mL; 24, 48 h	① Suppression of proliferation ② Inhibition of metastasis and angiogenesis		[48]
		In vivo	zebrafish	0.3, 0.6 mg/mL; 3 days			
Single Extract	Whole plant of *Viola yedoensis* (ethanol)	In vitro	A549, LLC	10, 25, 50, 75, 100 µg/mL; 24 h	① Inhibition of migration and invasion ② Inhibition of metastasis	↓: MMP-2, 9, uPA, TIMP-1, 2, NF-κB, p-p38	[49]
Mixture Extract	BushenShugan Formula	In vitro	A549	1.25, 3.75 g/mL; 24, 48, 72 h	① Suppression of proliferation ② Induction of apoptosis ③ Inhibition of invasion and metastasis ④ Inhibition of EMT ⑤ Decrease in colony formation ⑥ Blockage of CSC properties	↑: Bax, p53, E-cadherin ↓: Bcl-2, cyclin D1, N-cadherin, vimentin, α-SMA, Snail, P-PI3K, P-AKT, P-NF-κB	[50]
		In vivo	SD rats	30 g/kg; 7 days			
Mixture Extract	Feiji	In vitro	2LL-EFGP, 2LL-EFGP-IDO	5, 10, 15, 20%; 72 h	① Regulation of immune response. ② Suppression of tumor progression ③ Prolongation of survival time ④ Induction of apoptosis	↑: tryptophan ↓: IDO, Tregs, kynurenine	[51]
		In vivo	2LL-EFGP-IDO- xenografted C57BL/6 mice	5, 10, 15, 20%; 4, 8, 12 days			
Mixture Extract	Fei-Liu-Ping	In vivo	LLC- xenografted C57BL/6 mice	2 g/mL; 14, 21 days	① Inhibition of tumor growth ② Inhibition of metastasis.	↓: COX2, N-cadherin, MMP-9, vimentin	[52]
Mixture Extract	Fuzheng Kang-Ai	In vitro	PC9, A549, H1650	1, 1.2, 2, 3 mg/mL; 12, 24 h	① Suppression of proliferation ② Inhibition of migration and invasion	↓: STAT3/MMP-9, N-cadherin, vimentin	[53]
Mixture Extract	Jinfu’an	In vitro	H1650	2, 4, 6 mg/mL; 8, 12 h	① Suppression of proliferation ② Inhibition of adhesion, migration, invasion and metastasis	↑: Kaiso ↓: p120ctn, isoform 1A, psS288	[54]
Mixture Extract	Jinfukang	In vitro	LEC	1.8, 3.6, 7.2 g/kg; 3 days	① Suppression of formation ② Inhibition of migration ③ Inhibition of tumor lymphangiogenesis	↓: LECs, VEGFC, VEGFR3, SDF-1, CXCR4	[55]
Mixture Extract	Jingfukang	In vitro	CTC-TJH-01	350, 700 µg/mL; 48 h	① Suppression of proliferation ② Induction of apoptosis ③ Inhibition of metastasis ④ Decrease in colony formation ⑤ Arrest of cell cycle (G1)	↑: caspase-3, ROS, γ-H2AX, p-ATM, p-ATR, PARP1, p53, p21, Fas ↓: survivin, CDK2, 4, 6, cyclin D1, cyclin E1	[56]
Mixture Extract	JP-1	In vitro	A549	100, 150, 200, 250, 300, 350, 400, 450 µg/mL; 48 h	① Suppression of proliferation ② Inhibition of adhesion and migration ③ Decrease in colony formation	↑: AMPK, p53, p21, Bax, PARP, caspase-3, 9, miR-34a ↓: CDK6, p-pRb, p-mTOR, AXL, Snail, vimentin, SIRT1, c-Myc, survivin	[57]
Mixture Extract	MA128	In vitro	HT1080	50, 1000 µg/mL; 48 h	① Suppression of cell viability and tumor growth ② Inhibition of adhesion, migration, invasion, metastasis and angiogenesis ③ Decrease in colony formation	↑: p21, p27, p38, Bad, Bax, ERK ↓: cyclin B, cyclin D, MMP-9, NF-κB, Bcl-2, XIAP	[58]
		In vivo	HT1080-inoculated athymic nude mice	75, 150 mg/kg; 12 days			
		In vivo	B16F10-injected C57BL/6 mice	75, 150 mg/kg; 17 days			
Mixture Extract	Qingzaojiufei	In vitro	LLC	5%, 10%, 20%; 24 h	① Suppression of proliferation and tumor growth ② Induction of apoptosis ③ Inhibition of invasion and metastasis	↑: p53 ↓: PCNA, MMP-9, VEGF/VEGFR, c-Myc, Bcl-2, p-ERK1/2	[59]
		In vivo	LLC- xenografted C57BL/6 mice	3.8, 7.6, 1.52 g/kg; 4 weeks			
Mixture Extract	Xiaoai Jiedu Recipe	In vitro	A549	3.78, 7.56, 15.12 g/kg; 24, 48, 72 h, 7–12 days	① Suppression of proliferation ② Induction of apoptosis ③ Inhibition of migration and invasion	↑: Bax, c-caspase-3, 9 ↓: MMP-2, 9, Bcl-2, p-p38, p-ERK, p-JNK	[60]
		In vivo	SD rats	3.28, 7.56, 15.12 g/kg; 2 days			
Mixture Extract	Xiaoji	In vitro	A549, PC9	10, 20, 30, 40, 50 mg/mL; 24 h	① Suppression of tumor growth	↑: AMPKα ↓: DNTM1, Sp1	[61]
		In vivo	xenograft nude mice	6.7, 13.4, 26.8 g/kg; 30 days			

HUVEC, human umbilical vein endothelial cell; LLC, Lewis lung carcinoma cells; EFGP, enhanced green fluorescent protein; IDO, indoleamine-2,3-dioxygenase; LEC, lymphatic endothelial cell; CTC, circulating tumor cell; P-VEGFR2, phosphorylated VEGFR2; VEGER, vascular endothelial growth factor receptor; PLCγ1, phospholipase C gamma 1; FAK, focal adhesion kinase; AKT, serine/threonine protein kinase; STAT3, signal transducer and activator of transcription 3; JAK2, janus kinase 2; CD31, cluster of differentiation 31; MMP, matrix metalloproteinase; uPA, urokinase-type plasminogen activator; NF-κB, nuclear factor-kappa B; EGF, Epidermal Growth Factor; Ang-2, Angiopoietin-2; PDGF, Platelet-derived growth factor; IL-8, Interleukin 8; VEGF, vascular endothelial growth factor; PIGF, Placental Growth Factor; TIMP, tissue inhibitors of metalloproteinase; Bax; Bcl-2 associated X protein; Bcl-2, B-cell lymphoma 2; α-SMA, alpha smooth muscle actin; PI3K, phosphatidylinositol-3-kinase; Treg, regulatory T cell; COX2, cyclooxygenase 2; ps288, S288 phosphorylation; ROS, Reactive oxygen species; γ-H2AX, phosphorylated H2A histone family member X; ATM, ataxia telangiectasia mutated; ATR, ATM and Rad3-related; PARP1, poly (ADP-ribose) polymerase 1; survivin, surviving protein; CDK; cyclin-dependent kinase; AMPK, AMP-activated protein kinase; AMP, adenosine monophosphate; miR-34a, MicroRNA 34a; mTOR, mammalian target of rapamycin; SIRT1, sirtuin 1; ERK, extracellular regulated protein kinase; XIAP, X-linked inhibitor of apoptosis protein; PCNA, proliferating cell nuclear antigen; JNK, c-Jun N-terminal kinase; DNMT1, DNA methyltransferase 1.

**Table 3 antioxidants-10-00527-t003:** Colorectal cancer.

Classification	Herbal Medicine	System	Experimental Model	Dose; Duration	Efficacy	Mechanism	Reference
Single Extract	*Arctii Fructus* (water, ethanol)	In vitro	CT26	1, 10, 100 μg/mL; 24, 48 h	① Suppression of proliferation ② Inhibition of migration, invasion and metastasis ③ Induction of apoptosis ④ Arrest of cell cycle (G_2_/M)	↑: c-caspase-3, 8, 9, pro-PARP, E-cadherin, AMPK ↓: Bcl-2, Bcl-xL, N-cadherin, MMP-2, 9	[63]
Single Extract	*Scutellaria barbata* (water)	In vitro	HCT116	50–800 μg/mL; 48 h	① Suppression of proliferation, viability and tumor growth ② Inhibition of migration, invasion and metastasis ③ Inhibition of lung metastasis	↑: caspase-3, PARP, p53, E-cadherin ↓: CXCR4, Tspan 8, ERK, β-catenin, Src	[64]
		In vivo	HCT116-inoculated nude mice	615, 1230 mg/kg; 4 weeks			
		In vivo	colon26 tumors-inoculated BALB/c mice	615, 1230 mg/kg; 4 weeks			
Single Extract	Sprout of *Coix lacryma-jobi var. ma-yuen* Stapf (water)	In vitro	HCT116, HUVECs	0.5, 1 mg/mL; 8, 24, 48, 72 h	① Suppression of proliferation and tube formation ② Inhibition of adhesion, migration and invasion	↓: ERK1/2, AKT	[65]
Single Extract	aRVS combined with Dokhwaljihwang-tang	Clinical study	lung metastasis from rectal cancer patient	1.350 g/day (aRVS) + 300 mL/day (Dokhwaljihwang-tang) ; 2 years	① Inhibition of lung metastasis ② Improvement of progression-free survival		[66]
Mixture Extract	*Astragalus Atractylodes* mixture	In vitro	HCT-116, LoVo	0.5, 1, 2, 4, 6, 8, 12, 16 mg/mL; 24 h	① Inhibition of migration and invasion ② Inhibition of the lung metastasis ③ Suppression of VM formation	↓: ROS, HIF-1a, MMP-2	[67]
				2.5 mg/mL; 24 h			
		In vivo	HCT-116 inoculated athymic nude mice	16 mg/g; 50 days			
Mixture Extract	Dahuang Zhechong Pill	In vivo	MC38-EGFP inoculated C57BL/6J mice	9.6 mg/10 g/d	① Inhibition of liver metastasis ② Reduction in the metastatic tumor number	↑: cisplatin ↓: TGF-β1, CCL2, CCR2, F4/80, FN	[68]
Mixture Extract	Danggui-Sayuk-Ga-Osuyu-Senggang-Tang	In vitro	HCT116	50 µg/mL; 2 h	① Inhibition of migration and invasion	↑: E-cadherin, ZO-1 ↓: AKT, GSK-3β, N-cadherin, claudin-1	[69]
Mixture Extract	Formula of the aqueous extracts of 18 herbs	Clinical study	mCRC patients	Individualized dose; more than 2 months	① Prolongation of median survival time ② Inhibition of migration ③ Suppression of proliferation	↓: ErbB2, PPARγ, RXR, VEGFR, VEGFA, PI3K/AKT, Src, TNF-α	[70]
		In vitro	HT29	100, 200, 300, 400 μg/mL; 12, 36 h			
Mixture Extract	Janpi Jiedu	Clinical study	CRC patients	200, 400 mL; more than 3 months	① Improvement of survival rate		[71]
Mixture Extract	Janpi Jiedu	In vitro	HCT116, HT29, LoVo, SW48	0.3125, 0.625, 1.25, 2.5 mg/mL; 24, 48, 72 h	① Suppression of proliferation, viability and tumor growth ② Inhibition of migration, invasion, angiogenesis and metastasis ③ Induction of apoptosis	↓: CD34, VEGF, mTOR, HIF-1α, p70S6K, 4E-BP1, p-p70S6K	[72]
		In vitro	HCT116	0.2, 0.4, 0.8 mg/mL; 72 h			
		In vivo	HCT116- inoculated BALB/c nude mice	2 g/mL; 20 days			
Mixture Extract	Janpi Jiedu	In vitro	LoVo	12.5, 25, 50 μg/mL; 48 h	① Inhibition of liver and lung metastasis ② Inhibition of migration and invasion ③ Prolongation of survival time	↑: E-cadherin, Smad2/3 ↓: TGF-β/Smad, Vimentin, p-Smad2/3, Snail, MMP-2, 9	[73]
		In vivo	BALB/c nude mice inoculated LoVo	250, 500, 1000 mg/kg/day; 28 days			
Mixture Extract	Modified Anti-cancer Decoction II Formula	Clinical study	CRC patients	300 mL; more than 6 months	① Prolongation of survival time ② Inhibition of migration and invasion		[74]
Mixture Extract	Modified Si-Jun-Zi Decoction	In vivo	GFP-HCT-116- inoculated BALB/c nude mice	45 g/kg; 3 weeks	① Inhibition of liver metastasis ② Prolongation of survival time	↑: GM-CSF, IFN-γ, IL-1α, IL-3	[75]
Mixture Extract	Onbaekwon	In vitro	HCT116, MDA-MB-231, MCF7, HepG2, Hep3B	0.8 mg/mL; 24 h	① Inhibition of migration, invasion and metastasis	↓: CXCR4, CXCL12, NF-κB	[76]
Mixture Extract	Pien Tze Huang	In vitro	CT-26	0.25, 0.5, 0.75 mg/mL; 24, 48, 72 h	① Inhibition of migration, invasion and metastasis ② Inhibition of liver metastasis ③ Suppression of proliferation, viability and tumor growth	↑: E-cadherin ↓: TGF-β1, Smad2/3, 4, N-cadherin	[77]
		In vivo	BALB/c mice	234 mg/kg/day; 14 days			
Mixture Extract	Pien Tze Huang	In vitro	HCT-8	0.25, 0.5, 0.75 mg/mL; 24 h	① Inhibition of migration, invasion and metastasis	↑: E-cadherin, miR-200a/b/c ↓: TGF-β1, Smad2/3, 4, ZEB-1, 2, N-cadherin	[78]
Mixture Extract	Pien Tze Huang	In vitro	HCT-8, HCT-116, SW620	0.25, 0.5, 0.75 mg/mL; 24, 48 h	① Inhibition of migration and metastasis ② Suppression of lymph-angiogenesis ③ Suppression of viability and tube formation	↓: VEGFR3, VEGFC, MMP-2, 9	[79]
Mixture Extract	Quxie Capsule	Clinical study	mCRC patients	50 mg/kg; 3 months	① Prolongation of survival time		[80]
Mixture Extract	Teng-Long-Bu-Zhong-Tang	In vivo	RKO inoculated BALB/c nude mice	11.25, 22.5 g/kg/0.3 mL; 2 months	① Inhibition of lung metastasis	↓: LOX, HIF-1α, p-FAK, integrin αVβ3	[81]
Mixture Extract	Weichang’an	In vitro	HCT-116	3, 6, 9%; 24, 48, 72 h (decoction)	① Suppression of proliferation, viability and tumor growth ② Inhibition of liver metastasis	↓: β-catenin, MMP-7, CEA	[82]
				5, 10, 20%; 24, 48, 72 h (serum)			
		In vivo	HCT-116 inoculated BALB/c nude mice	0.5 mL/d; 7 days			
Mixture Extract	Xiaotan Tongfu	In vivo	CT26 inoculated BALB/c mice	3.536 mg/(g.d); 4 weeks	① Inhibition of liver metastasis	↑: IGFBP-3 ↓: IGF-1, IGF-1R	[83]
Mixture Extract	Zuo Jin Wan	In vitro	SW403	25, 50, 100 μg/mL; 24 h	① Arrest of cell cycle (G1) ② Inhibition of migration, invasion and metastasis	↓: 5-HTR1D, β-catenin, CDK4, cyclin D1, c-Myc, MMP-2, 7, CXCR4, ICAM-1	[84]
Mixture Extract	individualized TCM decoction	Clinical study	CRC patients	Individualized dose; more than one year	① Improvement of overall survival ② Improvement of disease-free survival		[85]

aRVS, allergen-removed Rhus verniciflua stokes; TCM, traditional Chinese medicine; c-caspase, cleaved caspase; PARP, poly ADP ribose phosphorylase; E-cadherin, epithelial cadherin; AMPK, AMP-activated protein kinase; Bcl-2, B-cell lymphoma-2; Bcl-xL, B-cell lymphoma-extra large; N-cadherin, neural-cadherin; MMP, matrix metalloproteinase; ERK, Extracellular signal-regulated kinase; AKT, Serine-threonine kinase; ROS, Reactive oxygen species; HIF-1α, hypoxia-inducible factor 1α; cisplatin, cis-dichlorodiammineplatinum-II; TGF-β1, transforming growth factor beta 1; CCL2, C-C motif chemokine ligand 2; CCR2, C-C motif chemokine receptor 2; FN, fibronectin; ZO-1, zonula occludens-1; GSK-3β, glycogen synthase kinase 3β; ErbB2, erythroblastic oncogene B-2 gene; PPARγ, peroxisome proliferator-activated receptor γ; RXR, retinoid X receptor; VEGFR, vascular endothelial growth factor receptor; VEGFA, vascular endothelial growth factor A; PI3K, phosphatidylinositol-3-kinase; TNF, tumor necrosis factor; CD34, Cluster of differentiation 34; VEGF, vascular endothelial growth factor; mTOR, mammalian target of rapamycin; p70S6K, ribosomal protein S6 kinase; 4E-BP1, eIF4E binding protein-1; Smad, mothers against decapentaplegic homolog; p-Smad2/3, phospho-Smad2/3; GM-CSF, granulocyte macrophage-colony stimulating factor; IFN-γ, interferon gamma; IL-1α, interleukin-1α; IL-3, interleukin-3; CXCR4, C-X-C chemokine receptor type 4; CXCL12, C-X-C chemokine ligand 12; NF-κB, nuclear factor-κB; ZEB, zinc finger E-box-binding homeobox; LOX, lysyl oxidase; p-FAK, phosphorylation of focal adhesion kinase; CEA, carcinoembryonicantigen; IGFBP-3, IGF binding protein 3; IGF-1, insulin-like growth factor 1; IGF-1R, IGF-1 receptor; 5-HTR1D, 5-hydroxytryptamine receptor 1D; β-catenin, catenin beta-1; CDK4, cyclin-dependent kinase 4; ICAM-1, intercellular adhesion molecule-1.

**Table 4 antioxidants-10-00527-t004:** Gastric cancer.

Classification	Herbal Medicine	System	Experimental Model	Dose; Duration	Efficacy	Mechanism	Reference
Single extract	*Actinidia chinensis* Planch	In vitro	HGC-27	90, 180, 360 mg/mL; 24, 48 h	① Suppression of proliferation and tumor growth ② Induction of apoptosis and ferroptosis ③ Inhibition of migration and metastasis	↑: ROS ↓: Vimentin, N-cadherin, Snail, GPx4, xCT	[87]
		In vivo	zebrafish	90, 180 mg/mL; 48 h			
Single extract	Dried root of *Salvia miltiorrhiza* (water)	In vitro	BGC-823	15 mg/mL; 48 h	① Induction of apoptosis ② Inhibition of metastasis ③ Regulation of immune response	↓: NET, NOXs, MPO	[88]
		In vivo	BGC-823-RFP-inoculated BALB/c athymic nude mice	780 mg/kg; 4 weeks			
Mixture extract	Babao Dan	In vitro	AGS, MGC80-3	0.25, 0.5, 0.75 mg/mL; 12, 24, 36, 48, 72 h	① Suppression of proliferation and viability ② Inhibition of migration, invasion and metastasis	↑: E-cadherin ↓: N-cadherin, vimentin, ZEB-1, 2, Twist1, MMP-2, 9, TGF-*β*1, p-Smad2/3	[89]
Mixture extract	Daikenchuto	In vitro	MCF7, KYSE790, DLD	20 μg/mL; 24, 48, 72 h	① Suppression of tumor growth ② Inhibition of migration, invasion ③ Induction of apoptosis		[90]
		In vitro	MKN45	0.2, 0.66, 2, 6.6, 20 μg/mL; 60 h			
		In vivo	nude mice	0.002 g/0.2 mL; 4 weeks			
Mixture extract	Jianpi Bushen	In vivo	MFC-inoculated strain 615 mice	20 g/kg; 7 days	① Inhibition of lung metastasis	↓: Rac1, Cdc42, SDF-1, FN	[91]
Mixture extract	Modified Jianpi Yangzheng	In vitro	HGC-27	1, 2 mg/mL; 24, 48 h	① Suppression of proliferation ② Inhibition of metastasis	↑: E-cadherin, p-AKT, p-IκKα/β, p-C/EBPβ, TNF-α, IL-1β, IL-12p ↓: N-cadherin, PI3Kγ, p-NF-κB, IL-10	[92]
		In vivo	strain 615 mice	10, 20 g/kg; 14, 30 days			
Mixture extract	Jinlong Capsule	In vitro	MGC-803, BGC-823	0.1, 0.2, 0.4, 0.8 mg/mL; 24 h	① Suppression of proliferation ② Induction of apoptosis ③ Arrest of cell cycle (S, G_2_/M)	↑: Bax, caspase-3, c-caspase-3 ↓: Bcl-2, survivin	[93]
		In vivo	BGC-823-RFP-xenografted BALB/c athymic nude mice	390, 780, 1560 mg/kg; 28 days			
Mixture extract	Xiao Tan He Wei	In vitro	MNNG induced GES-1	0.4 g/L; 24 h	① Suppression of proliferation ② Induction of apoptosis ③ Arrest of cell cycle (G0/G1)	↑: Bax, caspase-3, IkB ↓: Bcl-2, NF-κB, p65	[94]
		In vivo	PLGC rat	0.34 mL/100 g; 6 weeks			
Mixture extract	Xiaotan Sanjie	In vitro	HUVECs, SGC-7901	10.3 g/mL; 8, 12, 16, 24 h	① Suppression of tube formation ② Inhibition of migration and angiogenesis	↓: VEGFA, VEGFR1, VEGFR2	[95]
Mixture extract	ZiYinHuaTan	In vitro	HGC27, MGC803	50, 100, 200 μg/mL; 24, 48, 72 h	① Suppression of proliferation and tumor growth ② Induction of apoptosis ③ Arrest of cell cycle (G0/G1)	↓: PI3K, cyclin D1, Bcl-2, AKT	[96]
		In vivo	HGC27-inoculated BALB/c nude mice	940, 1880, 3760 mg/kg/day; 28 days			

RFP, red fluorescent protein; MFC, mouse forestomach carcinoma; MNNG, 1- methyl-3-nitro-1-nitrosoguanidine; PLGC, precancerous lesions of gastric carcinoma; HUVEC, human umbilical vein endothelial cell; ROS, reactive oxygen species; GPx4, glutathione peroxidase 4; xCT, cystine/glutamate antiporter; NET, neutrophil extracellular trap; NOX, NADPH oxidase; MPO, myeloperoxidase; ZEB, zinc finger E-box binding homeobox; MMP, matrix metalloproteinase; Rac-1, Ras-related C3 botulinum toxin substrate 1; Cdc42, cell division control protein 42 homolog; SDF, stromal cell-derived factor; FN, fibronectin; p-AKT, phospho-serine/threonine protein kinase; C/EBPβ, CCAAT/enhancer-binding protein-beta; TNF-α, tumor necrosis factor-alpha; IL-1β, interleukin 1 beta; PI3Kγ, phosphoinositide 3-kinase-gamma; NF-κB, nuclear factor-kappa B; Bax, Bcl-2 associated X protein; c-caspase-3, cleaved caspase-3; survivin, surviving protein; IkB, inhibitor of kB; VEGF, vascular endothelial growth factor; VEGFR, vascular endothelial growth factor receptor.

**Table 5 antioxidants-10-00527-t005:** Liver cancer.

Classification	Herbal Medicine	System	Experimental Model	Dose; Duration	Efficacy	Mechanism	Reference
Single extract	*Actinidia chinensis* Planch root (water)	In vitro	Hep3B, HepG2, SMMC7721, MHCC97L, MHCC97H, HCCLM3	10 mg/mL; 24, 48, 72 h	① Suppression of proliferation ② Decrease in colony formation ③ Inhibition of migration and invasion ④ Induction of apoptosis ⑤ Inhibition of EMT	↑: E-cadherin, ZO-1 ↓: DLX2, Snail, N-cadherin, vimentin, ZEB-1	[98]
		In vivo	HCCLM3-inoculated BALB/c mice	2 g/kg/day; 4 weeks	① Inhibition of EMT ② Suppression of weight loss ③ Suppression of lung and intrahepatic metastasis and ascites		
Single extract	*Oldenlandia diffusa* (water)	In vitro	Huh7, HepG2	200 mg/mL; 24, 48, 72 h	① Suppression of proliferation ② Induction of apoptosis ③ Inhibition of migration	↑: caspase-3 ↓: ki67, CXCR1, CXCR2, CXCR4,	[99]
		In vivo	DEN-injected mice	100, 200 mg/kg; 28, 60 days	① Prolongation of survival time ② Suppression of proliferation ③ Inhibition of metastasis ④ Enhancement of liver function and glucose metabolism	↑: F-FDG ↓: ALP, AST, ALT,	
Single extract	*Rhazya stricta* (ethanol 70%, water 30%)	In vitro	HepG2	10, 20, 30 μg/mL; 12, 24, 48 h	① Suppression of proliferation and tube formation ② Inhibition of migration ③ Arrest of cell cycle (G2/M, G1/S)	↓: Cdc2, cyclin A1, cyclin B1	[100]
Single extract	*Salvia chinensis* Benth (water)	In vitro	PLC/ PRF/5	200, 250, 500 μg/mL; 24, 48, 72 h	① Suppression of proliferation and viability ② Inhibition of lung metastasis ③ Arrest of cell cycle (G0/G1)	↓: β-catenin, GSK3β	[101]
		In vitro	MHCC97L	300, 600, 1000 μg/mL; 24, 48, 72 h			
		In vivo	MHCC97L- implanted- athymic nude mice	100 mg/kg/d; 5 weeks			
Mixture extract	Buyang Huanwu	In vivo	HCCLM3-inoculated athymic BALB/c nude mice	0.2 mL; 21, 28, 35 days	① Inhibition of angiogenesis and metastasis	↑: VEGF ↓: RGS-5, HIF-1α	[102]
Mixture extract	Combination of *curcuma zedoary* and kelp	In vivo	H22-bearing mice	1000, 2000, 4000 mg/kg; 7 days	① Suppression of tumor growth ② Inhibition of metastasis	↓: MMPs, VEGF, p-AKT, p-ERK1/2, CBS, H2S	[103]
Mixture extract	Formula	Clinical study	hepatocarcinoma patient	2 years	① Regression of tumor mass ② Inhibition of omentum metastasis ③ Prolongation of survival time	↓: AFP	[104]
Mixture extract	Jie-du granule	Clinical study	BCLC-C stage HCC patients	8 g; 3 years	① Prolongation of survival time		[105]
Mixture extract	Sini-san	In vitro	HepG2-HBx	400, 800 μg/mL; 24 h	① Inhibition of migration, invasion and metastasis	↓: MMP-9, AP-1, ERK, JNK, NF-κB, IκB, PI3K/AKT	[106]
Mixture extract	SongYouYin	In vivo	Hepa1-6-inoculated C57BL/6 mice	0.2 mL; 42 days	① Decrease in tumor weight ② Inhibition of lung metastasis ③ Prolongation of survival time	↑: CD4, 8 ↓: TGF-β1, CD4+, 25+, Foxp3+ Tregs, TIL	[107]
Mixture extract	Yanggan Jiedu Sanjie	In vitro	Bel-7402	100 μg/mL; 48 h	① Inhibition of adhesion, migration and invasion ② Reversion of TGF- β1 induced morphological change	↑: E-cadherin ↓: TGF-β1, N-cadherin, Smad3, Snail	[108]

E-cadherin, epithelial-cadherin (CDH1, Cadherin 1); ZO, zonula occludens; N-cadherin, neural cadherin; ZEB, zinc finger E-box binding homeobox; Ki67, a cellular marker for proliferation; CXCR, CXC motif chemokine receptor; CXCL, CXC motif chemokine ligand; F-FDG, 2-[18F]-fluoro-2-deoxy-D-glucose; ALP, alkaline phosphatase; AST, aspartate transaminase; ALT, alanine aminotransferase; Cdc2, cell division control protein 2 homolog; GSK3β, glycogen synthase kinase-3β; VEGF, vascular endothelial growth factor; RGS‑5, regulator of G protein signaling 5; HIF‑1 α, hypoxia‑inducible factor 1 α; MMPs, matrix metallopeptidases; p-AKT, phosphorylated- serine-threonine kinase; p-ERK, phosphorylated extracellular signal-regulated kinases; CBS, cystathionine beta synthase; H2S, hydrogen sulfide; AFP, α-fetoprotein; BCLC-C, Barcelona-clinic liver cancer- C(advanced) stage; AP-1, activator protein-1; ERK, extracellular signal-regulated kinases; JNK, Jun N-terminal kinase; NF-κB, nuclear factor kappa-B; IκB, inhibitor of κB; PI3K, phosphoinositide 3-kinases; AKT, serine-threonine kinase; TGF, transforming growth factor; Tregs, regulatory T cells; TIL, tumor-infiltrating lymphocyte; Rac1, Ras-related C3 botulinum toxin substrate 1.

**Table 6 antioxidants-10-00527-t006:** Breast cancer.

Classification	Herbal Medicine	System	Experimental Model	Dose; Duration	Efficacy	Mechanism	Reference
Single extract	*Alisma canaliculatum* (ethanol)	In vitro	MDA-MB-231	10, 20 μg/mL; 6, 12, 24 h	① Inhibition of migration and invasion	↓: CXCR3, CXCL10, IκB, p65/RelA, IKK, NF-κB	[112]
Single extract	*Ampelopsis japonica* (ethanol)	In vitro	MDA-MB-231	200, 400 μg/mL; 12, 24 h	① Suppression of cell viability ② Inhibition of migration and invasion	↑: TIMP-1, 2 ↓: MMP-2, 9	[113]
Single extract	*Camellia sinensis*	In vivo	4T1-inoculated BALB/c mice	0.05, 0.1, 0.2 mg/mL; 24 h	① Induction of apoptosis ② Decrease in tumor burden ③ Inhibition of lung and liver metastasis ④ Inhibition of migration and invasion	↑: caspase-3 ↓: ALP, MMP-2, 9	[114]
Single extract	*Centipeda minima* (ethanol)	In vitro	MDA-MB-231	2.5, 5, 7.5, 10, 20 μg/mL; 24, 48, 72 h	① Suppression of proliferation ② Decrease in colony formation ③ Induction of apoptosis ④ Inhibition of migration and invasion	↓: EGFR, PI3K/AKT/mTOR, NF-κB, STAT3, MMP-9	[115]
		In vivo	MDA-MB-231- xenografted mice	25, 50 mg/kg; 4 weeks	① Suppression of tumor growth		
Single extract	*Cirsium japonicum* var. maackii (ethanol)	In vitro	HUVEC, MDA-MB-231	25, 50, 100, 200 µg/mL; 24 h	① Suppression of proliferation and tube formation ② Inhibition of angiogenesis	↓: VEGF, p-AKT, p-ERK	[116]
Single extract	*Curcumae radix* (ethanol)	In vitro	MCF7	25, 40 µg/mL; 48 h	① Inhibition of migration and invasion	↓: CCR7, MMP-9, COX2, c-FOS, c-JUN, ER-a, Fsp1, TFF1, Ccl21, Ccl19	[117]
		In vivo	MMTV-PyVT 634 Mul	50 mg/kg; 68 days	① Prolongation of survival time ② Suppression of proliferation, ③ Decrease in tumor burden ④ Inhibition of lung metastasis		
Single extract	*Oldenlandia diffusa* (ethanol)	In vitro	MDA-MB-231, MDA-MB-453	20, 40 µg /mL; 12, 24, 48 h	① Suppression of proliferation ② Induction of apoptosis ③ Inhibition of migration, invasion and metastasis	↑: p-H2AX ↓: Cav-1, MMP-2, 9, N-cadherin, vimentin	[118]
		In vivo	breast cancer xenotransplantated zebrafish	20, 40 µg /mL; 48 h	① Inhibition of migration, invasion and metastasis	↓: Cav-1	
Single extract	*Smilax china* L. (ethanol)	In vitro	MDA-MB-231	25, 50 µg/mL; 24 h	① Suppression of proliferation ② Inhibition of migration, invasion and metastasis	↑: TIMP-1, 2 ↓: uPA, uPAR	[119]
Single extract	*Solanum nigrum* (water)	In vitro	MCF-7	50, 100, 200 µg/mL; 12, 24 h	① Induction of apoptosis ② Arrest of cell cycle (G2/M) ③ Changes in mitochondrial morphology (Fission) ④ Inhibition of EMT	↑: E-cadherin, caspase-3, ROS ↓: ZEB-1, N-cadherin, vimentin	[120]
Single extract	*Rheum palmatum* L. (ethanol)	In vitro	MDA-MB-231	25, 50, 100, 150 µg/mL; 24, 48 h	① Inhibition of migration, invasion and metastasis	↑: PAI-1 ↓: uPA, uPAR, MMP-2, 9, IkBα, p38, ERK, AKT, JNK	[121]
Single extract	*Toxicodendron vernicifluum* Stokes (water)	In vivo	4T1-inoculated BALB/c mice	50, 100, 200, 400 mg/kg; 20 days	① Decrease in tumor weight and volume ② Inhibition of lung metastasis ③ Induction of apoptosis ④ Suppression of proliferation ⑤ Inhibition of angiogenesis	↑: PAI-1 ↓: Ki67, PECAM-1, VEGF, MMP-2, TIMP-1, uPA, ICAM-1, VCAM-1	[122]
Mixture extract	Formula	In vivo	4T1-inoculated BALB/c mice	231, 770, 2310 mg/kg; 4 weeks	① Decrease in tumor weight ② Inhibition of lung, liver and bone metastasis ③ Regulation of immune response ④ Prolongation of survival time	↑: IL-2, IL-12, IFN-γ, TNF-α ↓: CD4+, 25+, FOXP3+, CD11b+, Ly6G+	[123]
Mixture extract	Gubenyiliu II	In vitro	MCF-7, 4T1	200, 400, 450, 600, 800 μg/mL; 24, 48 h	① Suppression of tumor growth ② Inhibition of metastasis and angiogenesis	↓: heparanase, FGF-2, VEGF, p-AKT, p-ERK, CD31	[124]
		In vivo	4T1-luc2 inoculated BALB/c mice	5.0 g/kg/d; 25 days			
Mixture extract	Gubenyiliu II DR2 (huoxuehuayu)	In vitro	4T1	200, 400 μg/mL; 36, 48 h	① Suppression of proliferation ② Inhibition of metastasis and angiogenesis	↓: heparanase, FGF-2, VEGF, p-ERK, p-AKT	[124]
Mixture extract	Gubenyiliu II DR3 (jiedusanjie)	In vitro	4T1	200, 400 μg/mL; 36, 48 h	① Suppression of proliferation ② Inhibition of metastasis and angiogenesis	↓: heparanase, FGF-2, VEGF, p-ERK, p-AKT	[124]
Mixture extract	Ruyiping	In vitro	MDA-MB-231, MDA-MB-468	40%; 24, 48 h	① Suppression of tumor growth ② Inhibition of migration and invasion ③ Arrest of cell cycle (G2)	↑: E-cadherin ↓: CDK1, cyclin B1, MMP-9, N-cadherin, Vimentin, Snail1, Snail2	[125]
Mixture extract	Ruyiping combined with *Platycodon grandiflorum*	In vivo	4T1 inoculated BALB/c mice	5.67, 22.68 g/kg/d; 14 days	① Inhibition of lung metastasis via microenvironment modulation	↓: S100A8/A9, IL-1β, IL-6, CXCL2, 5, fibrinogen	[126]
Mixture extract	*Trametes robiniophila Murr* with *Radix Isatidis*	In vitro	SK-BR-3, MDA-MB-231	4, 6 mg/mL; 24, 48, 72 h	① Suppression of proliferation ② Arrest of cell cycle (G2/M) ③ Inhibition of migration and invasion ④ Induction of apoptosis	↑: p53, caspase-3 ↓: MMP-2, 9, Snail	[127]
Mixture extract	Wensheng Zhuanggu Formula	In vitro	MCF-7, MDA-MB-231, MDA-MB-231BO	5, 10, 20, 40 μg/mL; 24, 48 h	① Inhibition of BMSC-induced EMT ② Inhibition of invasion and migration ③ Inhibition of bone metastasis ④ Induction of apoptosis ⑤ Suppression of proliferation	↑: E-cadherin, occludin ↓: TGF-β1, Smads, N-cadherin, vimentin, Snail, Twist, Slug	[128]
		In vivo	BMSC/MDA-MB-231BO inoculated BALB/c nude mice	0.4, 0.8, 1.6 g/kg/d; 4 weeks			
Mixture extract	XIAOPI formula	In vitro	MDA-MB-231, MCF-7	600, 800, 1000 μg/mL; 24, 48, 72 h	① Little inhibitory effects		[129]
		In vivo	MMTV-PyMT+/− transgenic mice	0.5 g/kg/d; 55 days	① Suppression of tumorigenesis via micro- environment modulation ② Inhibition of lung metastasis and cancer stem cells	↑: TIMP-1 ↓: CXCL1, 2, CCL2, β-catenin, G-CSF	

CXCR, CXC motif chemokine receptor; CXCL, CXC motif chemokine ligand; IκB, Inhibitor of κB; IKK, IκB kinase; NF-κB, nuclear factor kappa-B; TIMP, tissue inhibitor of metalloproteinase; MMP, matrix metallopeptidases; ALP, alkaline phosphatase; EGFR, epidermal growth factor receptor; PI3K, phosphoinositide 3-kinases; AKT, serine-threonine kinase; mTOR, mammalian target of rapamycin; STAT3, Signal transducer and activator of transcription 3; VEGF, Vascular endothelial growth factor; p-AKT, phosphorylated- serine-threonine kinase; *p*-ERK, phosphorylated extracellular signal-regulated kinases; CCR, C-C chemokine receptor type; COX2, cyclooxygenase-2; c-FOS, proto-oncogene FOS; c-JUN, Jun proto-oncogene; ER-a, estrogen receptor alpha; Fsp1, Fibroblast Specific Protein 1; TFF1, trefoil factor 1; CCL, chemokine (C-C motif) ligand; p-H2AX, damage marker no.AP0640; Cav-1, caveolin-1; N-cadherin, neural-cadherin; uPA, urokinase plasminogen activator; uPAR, uPA receptor; EMT, epithelial-to-mesenchymal transition; ROS, reactive oxygen species; ZEB, zinc finger E-box-binding homeobox; PAI, plasminogen activator inhibitor; JNK, Jun N-terminal kinase; PECAM, platelet endothelial cell adhesion molecule; ICAM, intercellular adhesion molecule; VCAM, vascular cell adhesion molecule; IL, Interleukin; IFN-γ, Interferon gamma; TNF-α, tumor necrosis factor-alpha; Treg, regulatory T cells; FOXP3+, forkhead box P3; MDSC, myeloid-derived suppressor cells; CD+, cluster of differentiation; Ly6G+, lymphocyte antigen 6 complex locus G6D; FGF-2, fibroblast growth factor; DR, decomposed recipe; CDK1, cyclin-dependent kinase 1; E-cadherin, epithelial-cadherin; S100A, S100 calcium-binding protein A; BMSC, bone marrow-derived mesenchymal stem cells; TGF-β1, transforming growth factor beta 1; Twist, twist family bHLH transcription factor; G-CSF, granulocyte colony- stimulating factor.

**Table 7 antioxidants-10-00527-t007:** Components of Mixture extracts.

Target Cancer	Mixture Extracts	Components
Lung	Bushenshugan Formula	Bupleurum, Schisandra, Angelica, Red peony root, Medlar, Dodder, Plantain, Raspberry, Astragalus, Epimedium, Houttuynia, Trichosanthes, Allium white, Prunella, *Ligustrumlucidum, Houttuyniacordata,* Polyporus, Gentian, centipede
	Feiji	*Astragalus mongholicus, Radix Glehniae, Liriope spicata, Asparagus cochinchinensis, Poria cocos, Selaginella doederleinii, Salvia chinensia, Houttuynia cordata*, *Paris polyphylla*
	Fei-Liu-Ping	roots of *Astragalus membranaceus, Bge.var.mongholicus Hsiao, Panax quinquefolium L., Ophiopogon japonicas Ker-Gawl., Glehnia littoralis Fr. Schmidt ex Miq., Agrimonia pilosa Ledeb., Polygonum bistorta L., Patrinia villosa Juss., Panax notoginseng F.H. Chen, Fritillaria cirrhosa D.Don, Glycyrrhiza uralensis Fisch., Cordycrps sinensis Sacc.,* fruits of *Prunus persica Batsch, Prunus armeniaca L. var ansu Maxim*
	Fuzheng Kang-Ai	Taizishen*, Atractylodes macrocephala, Astragalus membranaceus, Oldenlandia diffusa, Solanum nigrum* L.*, Salvia chinensis, Cremastra appendiculata, Coix lachrymal-jobi* L.*, Akebia quinate, Rubus parviflolius* L.*, Curcuma kwangsiensis, Glycyrrhiza uralensis*
	Jinfu’an	*Pseudostellaria heterophylla, Cremastra appendiculata Makino, Coix lachrymaljobi* L. *Salvia miltiorrhiza Bunge,* gecko, peach kernel*, Phragmites communis,* Thunberg fritillary bulb, raw *Pinellia ternate,* raw *Rhizoma arisaematis*
	Jingfukang	*Astragalus membranaceus, Glehnia littoralis, Asparagus cochinchinensis, Ligustrum lucidum, Selaginella doederleinii, Paris polyphylla, Epimedium sagittatum, Gynostemma pentaphyllum, Cornus officinalis, Salvia chinensis, Ophiopogon japonicus, Trigonella foenum graecum*
	JP-1	*Ganoderma lucidum, Herba Scutellaria barbata, Scutellaria baicalensis, Oldenlandia diffusa, Astragalus membranaceus, Codonopsis Pilosula, Bulbus fritillariae cirrhosae,* and so forth.
	MA128	*Glycyrrhizae radix, Polygoni cuspidati radix, Sophorae radix, Cnidii rhizoma, Arctii fructus*
	Qingzaojiufei	frost mulberry leaves, plaster stone, baked licorice*, Codonopsis pilosula* root, donkey hide gelatin, dwarf lilyturf tuber, bitter almond, loquat leaves
	Xiaoai Jiedu Recipe	*Hedyotis diffusa, Cremastra appendiculata, Bombyx batryticatus, Centipede, Akebia trifoliata Koidz, Radix pseudostellariae, Radix ophiopogonis*
	Xiaoji Decoction	*Psoralea corylifolia Linn., Coriolus versicolor, Psoralea corylifolia Linn.,* Curcuma, *Hedyotis diffusa* Willd*., Buthus martensii* Karsch*, Scolopendra subspinipes, Rheum palmatum* L.
Colorectal	Dokhwaljihwang-tang	*Rehmanniae Radix Preparata, Corni Fructus, Hoelen, Alismatis Rhizoma, Moutan Cortex Radicis, Saposhnikoviae Radix, Araliae Continentalis Radix*
	*Astragalus Atractylodes* mixture	*Astragalus membranaceus Fisch, Atractylodes macrocephala Koidz, Actinidia arguta Planch, Curcuma aromatica Salisb, Benincasa hispida Cogn., Ficus pumila L*
	Dahuang Zhechong Pill	*Rehmanniae radix, Paeoniae radix alba, Glycyrrhizae radix, Rhei radix, Scutellariae radix, persicae semen, Armeniacae semen amarum, Tabanus, Hirudo, Holotrichae larva, Toxicodendri resina, Eupolyphaga seu steleophaga*
	Danggui-Sayuk-Ga-Osuyu-Senggang-Tang	Angelica radix, Cinnamomi cortex, Paeoniae root, Akebia root, Asarum, Glycyrrhiza*, Zizyphus jujube,* Evodia fruit, ginger root
	Formula of the aqueous extracts of 18 herbs	*Lycii Fructus, Magnolia ofcinalis Rehd Et Wils, Radix Clematidis, Aucklandiae Radix, Angelicae sinensis Radix, Xanthii Fructus, Eriocauli Flos, Cassiae Semen, Fallopia multifora, Selaginella doederleinii Hieron, Herba Patriniae, Portulacae Herba, Coicis Semen, Taraxacum mongolicum Hand, Agrimonia eupatoria, Ranunculi ternati Radix, Schisandrae chinensis Fructus, Radix Paeoniae rubra*
	Janpi Jiedu [71]	*Radix Astragali, Radix Ginseng, Rhizoma Atractylodis Macrocephalae, Poria Semen Coicis, Rhizoma Smilacis Chinensis, Herba Hedyotidis Diffusae, Herba Scutellariae Barbatae, Rhizoma Paridis, Radix Actinidiae Chinensis* (+Individualized adjustments)
	Janpi Jiedu [72]	*Astragalus propinquus, Panax quinquefolius, Atractylodesmacrocephala, Poria cocos,* Wolf*, Coix aquatica, Smilaxaberrans, Oldenlandia diffusa, Scutellaria barbata, Paris incompleta, Actinidiaacuminata, Glycyrrhizauralensis*
	Janpi Jiedu [73]	Radix Astragal, Rhizoma Atractylodis Macrocephala, wild grapevines, Fructus Akebia, *Salvia chinensis* Benth*, Evodia rutaecarpa*
	Modified Anti-cancer Decoction II Formula	Radix Astragali, Rhizoma Atractylodis, Codonopsis Pilosula, Poria Cocos, Tangerine Peel, Semen Coicis, Dioscorea Opposita, Fructus Lycii, Glossy Privet Fruit, Angelica Sinensis, Rhizoma Polygonati, Oldenlandia Diffusa, Chinaroot Greenbrier, Wild Grape-vine, Radix Actinidiae Chinensis, Rehmannia Glutinosa Libosch, Selfheal, Gecko, Centipede
	Modified Si-Jun-Zi Decoction	*Radix Astragali, Codonopsis pilosula, Rhizoma Atractylodis Macrocephalae, Poria, Dioscoreae Rhizom, Glycyrrhizae Radix et Rhizoma, Mume Fructus, Sparganii Rhizoma Curcumae Rhizoma, Agrimoniae Herba, patrinia*
	Onbaekwon	*Aconitum carmichaelii, Evodia rutaecarpa,* Platycodi Radix, Bupleuri Radix, Acori Rhizoma, Asteris Radix, Coptidis Rhizoma, Zingiberis Rhizoma, Cinnamomi Cortex, Tiglii Fructus, Hoelen, *Gleditsia sinensis* Lam, Machili Cortex, Ginseng Radix, Zanthoxyli Fructus
	Pien Tze Huang	Moschus, Calculus Bovis, Snake Gall, Radix Notoginseng
	Quxie Capsule	*Croton tiglium, Evodia rutaecarpa, Rhizoma zingiberis, Cinnamomum cassia* Presl*, Radix aconiti, Pinellia ternate, Pericarpium Citri Reticulatae*
	Teng-Long-Bu-Zhong-Tang	*Actinidia chinensis, Solanum nigrum, Duchesnea indica, Atractylodes macrocephala, Coix* seed*, Viscum coloratum*
	Weichang’an	*Pseudostellaria heterophylla* Pax*, Atractylodes macrocephala* koidz.*, Poria cocos* Wolf*, Glycyrrhiza uralensis* Fisch.*, Sargentodoxa cuneata, Prunella vulgaris* L.
	Xiaotan Tongfu	*Pinelliae rhizome, Citrus reticulata Blanco, Poria*
	Zuo Jin Wan	*Rhizoma Coptidis, Evodia Rutaecarpa*
	Babao Dan	bezoar, snake gall, antelope horn, pearl, musk, *Panax notoginseng*
	Daikenchuto	viz., ginseng, Japanese Zanthoxylum peel, processed ginger
Gastric	Jianpi Bushen	*Radix Codonopsis, Fructus Lycii, Rhizoma Atractylodis Macrocephalae, Fructus Ligustri Lucidi, Cuscuta chinensis, Psoralea corylifolia Linn*
	Jianpi Yangzheng Xiaozheng	*Astragalus Root, Codonopsis Pilosula, Rhizoma Atractylodis Macrocephalae, Angelica Sinensis, Radix Paeoniae Alba, Rhizoma Sparganii, Curcuma Zedoary, Pericarpium Citri Reticulatae, Costustoot, Oldenlandia Diffusa, Salvia Chinensis, Raw licorice*
	Modified Jianpi Yangzheng	*Radix astragali, Radix codonopsis pilosulae, Rhizoma Sparganii, Rhizoma Curcumae*
	Jinlong Capsule	*Bungarus, Agkistrodon, Gecko*
	Xiao Tan He Wei	*Radix Bupleuri, Processed Rhizomapinelliae, Poria Cocos, Coptis chinensis, Oldenlandia diffusa, Dandelion, Cassia Twig, Rhubarb, Radix Paeoniae Alba, Radix Glycyrrhizae Preparata*
	Xiaotan Sanjie	*Pinelliae rhizome, Rhizoma arisaematis, Poria cocos, Aurantii fructus immaturus, Citri reticulatae viride pericarpium, Scorpio, Scolopendra, Galli gigerii endothelium corneum, Fritillariae cirrhosae bulbus, Semen brassicae, Glycyrrhiza uralensis Fisch*
	ZiYinHuaTan	*Lily, Pinellia, Hedyotis diffusa*
Liver	Buyang huanwu	milkvetch root*, Chinese angelica,* red peony root*,* earth worm*, Szechwan lovage rhizome,* peach seed*,* safflower
	Combination of *curcuma zedoary* and kelp	*Curcuma phaeocaulis, Curcuma kwangsensis* (or *Curcuma wenyujin), Laminaria japonica Aresch* (or *Ecklonia kurome)*
	Formula [104]	*Radix Bupleuri, Radix Paeoniae Alba, Angelica Sinensis, Radix Curcumae, Radix Salviae Miltiorrhizae, Ligusticum Wallichii, Actinidia valvata Dunn, Melia Toosendan, Fructus Aurantii, Artemisia carvifolia, Caulis Spatholobi, Chinese Lobelia, Sedum Sarmentosum Bunge, Squama Manis Radix Ranunculi Ternati, Salvia Chinensis, Hedyotis Diffusa, Liquidambar Formosana Hance Centipede*
	Sini-san	*Radix Bupleuri Chinensis, Radix Paeoniae Alba, Fructus Aurantii Imma Turus, Radix Glycyrrhizae*
	SongYouYin	*Salvia miltiorrhiza, Astragalus membranaceus, Lycium barbarum, Crataegus pinnatifida, Trionyx sinensis*
Breast	Formula [123]	*Andrographis paniculata, Acanthopanax senticosus, Camellia sinensis, Hedyotis diffusa*
	Gubenyiliu II	*Codonopsis pilosula, Poria cocos, Atractylodis macrocephala, Astragalus membranaceus, mongholicus, Ligustrum lucidum, Lycium barbarum, Epimedium brevicornum, Ligusticum chuanxiong, Spatholobus suberectus, Curcuma kwangsiensis, Sarcandra glabra, Fritillaria thunbergii, Sophora flavescens*
	Gubenyiliu II- DR2	*Ligusticum chuanxiong, Spatholobus suberectus, Curcuma kwangsiensis,*
	Gubenyiliu II-DR3	*Sarcandra glabra, Fritillaria thunbergii, Sophora flavescens*
	Ruyiping	*Pseudobulbus Cremastra, Nidus Vespae, Curcuma zedoaria,* raw seeds of *Coix lacryma-jobi Linne var. mayuen, Stapf, Akebiae Fructus*
	Ruyiping + *Platycodon grandiflorum*	*Iphigeniaindica kunth, Curcuma zedoary, Nidus vespae, Semen coicis, Akebia fruit*
	Wensheng Zhuanggu Formula	*Psoraleae Fructus, Cnidii Fructus, Aconiti Lateralis, Radix Praeparata*
	XIAOPI formula	*Epimedium Brevicornum, Cistanche Deserticola, Leonurus Heterophyllus, Salvia Miltiorrhiza, Curcuma Aromatica, Rhizoma Curcumae, Ligustrum Lucidum, Radix Polygoni Multiflori Preparata, Crassostrea Gigas, Carapax Trionycis*

## Data Availability

Not applicable.

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
