# Peer review of "Recent Advances in Anti-Metastatic Approaches of Herbal Medicines in 5 Major Cancers: From Traditional Medicine to Modern Drug Discovery"

_antioxidants, 2021, doi:10.3390/antiox10040527_

Round 1

Reviewer 1 Report

Despite the interesting subject matter, the work presented for review is very poorly presented. First, the title suggests that the authors attempted to describe the effects of all, or at least most, herbal medicines in selected cancers. Unfortunately, these are only a few manuscripts; what's more, they are not crucial for a given topic. The authors choice seems to be quite accidental and devoid of logical sense.
The authors did not mention here f.e. the very often used evening primrose extract, the action of which in the case of colorectal cancer has been described in numerous studies.
Of course, the number and multithreading of herbal medicines that are now used may make it difficult to include all informations about their acting in one publication. But this can be counteracted. One approach is to select compounds that presented the same or similar effecton molecular pathays or selection of the compounds derived from similar chemical groups. Of course then the manucript title should be changed. On the other hand, if authors would like to present the wide review, then they should add references to well-written and recommended reviews describing compounds that will only be mentioned in their manuscript. Unfortunately, the authors did not undertake such critical analysis at all. Thus as a result their presented paper when described individual selected medicaments without logical sense and correlation.
The order of selected papers description is also quite random. There is no clearly defined pattern of descriptions here. Moreover, the schemas and figures that should support the content and make it easier to understand are so extensive that they only make the manuscript difficult for the reader to analyze the topic, making it appear disordered and quite convoluted.
Summing up, despite the fact that the work deals with a very important subject, the method of selecting materials, their analysis, and presentation is only a collection of information. It is not a review paper. I think that the authors should rethink the meaning of the publication and rewrite it so that it is a valuable element of an interesting and important research subject, and not another manuscript that will bring the authors only the value of scientific achievements without contributing anything to science.

Author Response

We appreciate editors and reviewers for giving us an opportunity to resubmit our manuscript (antioxidants-1087300), entitled “Effect of herbal medicines on cancer metastasis: A Review”. We earnestly responded to the raised comments point by point.

Despite the interesting subject matter, the work presented for review is very poorly presented. First, the title suggests that the authors attempted to describe the effects of all, or at least most, herbal medicines in selected cancers. Unfortunately, these are only a few manuscripts; what's more, they are not crucial for a given topic. The authors choice seems to be quite accidental and devoid of logical sense.
The authors did not mention here f.e. the very often used evening primrose extract, the action of which in the case of colorectal cancer has been described in numerous studies. Of course, the number and multithreading of herbal medicines that are now used may make it difficult to include all informations about their acting in one publication. But this can be counteracted.

(Response): We have clarified the appropriate selection criteria of article indicating the anti-metastatic effects of herbal medicine in the last five years, and we think it is not very appropriate to point out that the paper was selected randomly. As your comments, the fact that you think some of the articles have not been included may be that they were not appropriate in criteria of ‘herbal medicine + metastatic cancer’ that we set as keywords. Nevertheless, we double-screened 'evening primrose' to look at the deficiencies we might have missed, and confirmed that there is no article suitable for the selection criteria in our study. In other words, there have been many research using ‘evening primrose’ for cancer treatments as you said, but most of them were using extracted certain compounds such as flavanol, polyphenol, and procyanidins. Also, the other research is non-five major cancers, such as melanoma and prostate cancer etc. As described in methods parts, specific compounds have been excluded according to clear criteria in our study. Also, as you mentioned, we were able to check many articles using evening primrose have been reported in colorectal cancer, but most of them were extracted specific compound called polyphenol and sterol. In other words, there were no study using a single extract of evening primrose itself. By the way, we added two other articles during the more detailed cross-search process which had been omitted before. They are ‘Rhazya stricta’ in liver cancer and ‘Xiaoji decoction’ in lung cancer.

One approach is to select compounds that presented the same or similar effecton molecular pathays or selection of the compounds derived from similar chemical groups. Of course then the manucript title should be changed. On the other hand, if authors would like to present the wide review, then they should add references to well-written and recommended reviews describing compounds that will only be mentioned in their manuscript. Unfortunately, the authors did not undertake such critical analysis at all. Thus as a result their presented paper when described individual selected medicaments without logical sense and correlation. The order of selected papers description is also quite random. There is no clearly defined pattern of descriptions here.

(Response): Actually, we did consider the arrangement of papers according to similar effects or pathways. However, due to the nature of herbal medicine which is very complex and not limited to a single target, there were practical difficulties in ideal arrangement. Thus, in order to meet the goal of comprehensive analysis of the five major cancers, it was intended to be arranged by each of the five types of cancer and it was divided again into single and mixture extracts. After that, the studies were arranged in alphabetical order.

Moreover, the schemas and figures that should support the content and make it easier to understand are so extensive that they only make the manuscript difficult for the reader to analyze the topic, making it appear disordered and quite convoluted.

(Response): I don’t agree with your comments on figures. However, for better understating, we moved the existing description about signaling pathways which had been on the text part before to figure legends. As you know, there had been many overlapping descriptions of mechanisms in the <4. Results> text part. Hence, the content of text parts is simplified only to what is needed and general descriptions of mechanisms are transferred to figure legends so that these can improve reader’s understanding.

Summing up, despite the fact that the work deals with a very important subject, the method of selecting materials, their analysis, and presentation is only a collection of information. It is not a review paper. I think that the authors should rethink the meaning of the publication and rewrite it so that it is a valuable element of an interesting and important research subject, and not another manuscript that will bring the authors only the value of scientific achievements without contributing anything to science.

(Response): Since we reviewed 79 large articles, it might be seen that presentation is just collected information. However, these are not just the collections of data. We not only analyzed each existing article but also provided some comprehensive view of research which had been individually dispersed before. Also, from 79 papers, we also tried to grasp some tendencies and limitations of previous herbal medicine studies. Those are fully described in figures and discussion parts.

Again we appreciate reviewers for their kind and careful comments for improving the quality of our manuscript and also sincerely hope we address our responses well to the raised comments and our revised manuscript would be accepted for publication in Antioxidants journal soon.

Reviewer 2 Report

The present review entitled “Effect of herbal medicines on cancer metastasis: A Review” is an interesting work for researchers working in cancer metastasis and the role of herbal medicines in cancer research. Authors have reported different types of cancer and their pathways of interaction and mechanisms of possible herbal medicines. Mainly five types of cancers have been focused in this study based on their severe causes and concerns. This is comprehensive review summarizing from the 77 scientific articles, which shows good number of data sets. In this review paper, herbal mediacies have been used to investigate the anti-metastasis effect on major cancers type. This comprehensive review providing the experimental models, along with doses and treatment duration with possible mechanisms and pathways. Authors concluded that the herbal medicines could be a better option for anti-metastatic drugs by restricting several side effects. Authors have presented this review quite well and the message of the text is quite clear. A review in the area would be of great interest to the common readership of the journal.

Following suggestions could improve the manuscript-

  1. Methods section could be improved with more detail description about the article selection criteria, type of herbal medicines etc.
  2. Throughout the manuscript, herbal medicine has been used quite frequently. An inclusion of a table defining type of herbal medicines (extract/ commercial bioactive compounds or others) for all five types of cancers would be great inclusion. Maybe in vitro and in vivo based studies would be good inclusion.
  3. All figure captions should be extended with full description about the pathways and the mechanism. For readers, it will be useful to understand the mechanism.
  4. Authors should add a paragraph on the applications and future perspectives of herbal medicines in cancer related diseases and treatment possibilities after conclusion.

Author Response

We appreciate editors and reviewers for giving us an opportunity to resubmit our manuscript (antioxidants-1087300), entitled “Effect of herbal medicines on cancer metastasis: A Review”. We earnestly responded to the raised comments point by point.

The present review entitled “Effect of herbal medicines on cancer metastasis: A Review” is an interesting work for researchers working in cancer metastasis and the role of herbal medicines in cancer research. Authors have reported different types of cancer and their pathways of interaction and mechanisms of possible herbal medicines. Mainly five types of cancers have been focused in this study based on their severe causes and concerns. This is comprehensive review summarizing from the 77 scientific articles, which shows good number of data sets. In this review paper, herbal mediacies have been used to investigate the anti-metastasis effect on major cancers type. This comprehensive review providing the experimental models, along with doses and treatment duration with possible mechanisms and pathways. Authors concluded that the herbal medicines could be a better option for anti-metastatic drugs by restricting several side effects. Authors have presented this review quite well and the message of the text is quite clear. A review in the area would be of great interest to the common readership of the journal.

(Response): Thank you very much for your kind comments.

Following suggestions could improve the manuscript-

  1. Methods section could be improved with more detail description about the article selection criteria, type of herbal medicines etc.

(Response): We added more information about selection criteria in the Methods section.

  1. Throughout the manuscript, herbal medicine has been used quite frequently. An inclusion of a table defining type of herbal medicines (extract/ commercial bioactive compounds or others) for all five types of cancers would be great inclusion. Maybe in vitro and in vivo based studies would be good inclusion.

(Response): Thanks. Since we excluded the single compounds, we classified them as plant and decoction in a table defining type of herbal medicines before. But, as you advised, we changed the classification to single extract and mixture extract for an intuitive understanding. We also added criteria ‘system’ which includes ‘in vitro, in vivo or clinical study’.

  1. All figure captions should be extended with full description about the pathways and the mechanism. For readers, it will be useful to understand the mechanism.

(Response): We agree with the requirement for full description on figure captions. Thus, we moved the existing descriptions about pathways and mechanisms which had been on the <4. results>. Hence, the content of text parts is simplified only to what is needed and general descriptions of mechanisms are transferred to figure legends so that these can improve readers' understanding.

  1. Authors should add a paragraph on the applications and future perspectives of herbal medicines in cancer related diseases and treatment possibilities after conclusion.

(Response): We added sentences about the applications and future perspectives of herbal medicines at the end of conclusion.

Again we appreciate reviewers for their kind and careful comments for improving the quality of our manuscript and also sincerely hope we address our responses well to the raised comments and our revised manuscript would be accepted for publication in Antioxidants journal soon.

Reviewer 3 Report

The authors report data published in the last 5 years on antimetastatic potential of herbal medicine. The review has been written accurately and is well detailed.

- In paragraph 1.2 the authors should add some information on the efficacy of current drugs employed in metatstasis treatment’s approved from FDA

- In paragraph 2 the authors indicate the antioxidant properties as the main activitiy of herbal medicine, but should be discussed also the capacity of the complex matrix of compounds present in herbal medicine to modulate several signaling pathways involved in metastatic process

- At line 96 should be specified which herbal medicine has been studied in reference 26

- The manuscript is quite long. To shorten it slightly, I suggest reporting the results of the most interesting / representative papers in the first part of the several sub-paragraphs 4 (for example in 4.1 reduce the text from 144 to 224). For the other data collected it will be sufficient to refer to the tables and fugures, which are well done, clear and exhaustive, and well summarize the data collection made.

- The authors make good comments and observations on the data reported in the review. However, I suggest to further stress the limit of these studies, that is the wide variety of herbal medicines and the scarce experimental studies, which although sometimes promising do not provide sufficient support to demonstrate the real potential of the product.

Author Response

We appreciate editors and reviewers for giving us an opportunity to resubmit our manuscript (antioxidants-1087300), entitled “Effect of herbal medicines on cancer metastasis: A Review”. We earnestly responded to the raised comments point by point.

The authors report data published in the last 5 years on antimetastatic potential of herbal medicine. The review has been written accurately and is well detailed.

- In paragraph 1.2 the authors should add some information on the efficacy of current drugs employed in metatstasis treatment’s approved from FDA

(Response): In paragraph 1.2, we added some further efficacy information about employed drugs. Moreover, we thought that mentioning just two among numerus FDA-approved drugs is somewhat insufficient. Thus, we also added information about other recently approved drugs for metastasis, whose pathways are different from what we've mentioned before.

- In paragraph 2 the authors indicate the antioxidant properties as the main activitiy of herbal medicine, but should be discussed also the capacity of the complex matrix of compounds present in herbal medicine to modulate several signaling pathways involved in metastatic process

(Response): Thanks. Discussing the capacity of herbal medicine is very important as your comments. However, this part is included in introduction section. The signaling pathways regulated by herbal medicines were discussed in results and discussion section. We added more detailed signaling pathways in figure 6 and 7. Instead, we revised some sentences flow on paragraph 2 for better understanding.

- At line 96 should be specified which herbal medicine has been studied in reference 26

(Response): We added the specific name of herbal medicine in reference [28].

- The manuscript is quite long. To shorten it slightly, I suggest reporting the results of the most interesting / representative papers in the first part of the several sub-paragraphs 4 (for example in 4.1 reduce the text from 144 to 224). For the other data collected it will be sufficient to refer to the tables and fugures, which are well done, clear and exhaustive, and well summarize the data collection made.

(Response): Thank you for the comments. We shortened the manuscript form total 58 pages to 50 pages as your suggestion. However, reporting the result of the most interesting could be subjective. So, we reduced the detailed result and summarized the data.  

- The authors make good comments and observations on the data reported in the review. However, I suggest to further stress the limit of these studies, that is the wide variety of herbal medicines and the scarce experimental studies, which although sometimes promising do not provide sufficient support to demonstrate the real potential of the product.

(Response): You can say that again. You pointed the very important issue. That is why we have to study not only in non-clinical studies but also, in clinical studies. we added sentences to stress the limit of these studies that you advised in the part of discussion (text line 868-872). We hope this review article could be a significant database of actual clinical application.

Again we appreciate reviewers for their kind and careful comments for improving the quality of our manuscript and also sincerely hope we address our responses well to the raised comments and our revised manuscript would be accepted for publication in Antioxidants journal soon.

Reviewer 4 Report

The manuscript presents a very comprehensive compilation on studies focused on the use of herbal medicines to target metastasis. Although it is too long, it is well organized and contributes to provide an overall view of these type of therapies on cancer. 

Author Response

We appreciate editors and reviewers for giving us an opportunity to resubmit our manuscript (antioxidants-1087300), entitled “Effect of herbal medicines on cancer metastasis: A Review”. We earnestly responded to the raised comments point by point.

The manuscript presents a very comprehensive compilation on studies focused on the use of herbal medicines to target metastasis. Although it is too long, it is well organized and contributes to provide an overall view of these type of therapies on cancer.

(Response): Thank you for your kind comments. We shortened the manuscript form total 58 pages to 50 pages according to your suggestion.

Again we appreciate reviewers for their kind and careful comments for improving the quality of our manuscript and also sincerely hope we address our responses well to the raised comments and our revised manuscript would be accepted for publication in Antioxidants journal soon.

Round 2

Reviewer 1 Report

In the beginning, I would like to thanks the authors for the broad responses that could be convincing me to accept the presented manuscript.
However, in my opinion, the paper is still not consistent in several points. Despite the clarification of the manuscript title, it does not precisely indicate the contained issues. Moreover, it suggests that the authors carried out a much more in-depth analysis than is the case. The title should still be corrected.
Secondly, the narrowing of the topic is entirely incomprehensible. Why did the authors choose only works from the last five years? After all, they omitted some significant research and unconsciously mislead the reader, who, starting their interest in this important subject, may get the wrong impression that only these presented compounds, mixtures, extract were studied in the context of cancer metastasis. After all, review papers are not once an impulse to new research directions. Such limitations are wrong. However, if the authors still agree to their concept, they should refer to other, more broadly discussing the subject of review papers, encouraging the reader to refer to them. This aspect is very much missing here.
 Besides, even recognizing the authors' right to narrow down the subject, there is a lack of at least a few of the manuscripts that appear in Pubmed and meet the given criteria. Even if
Anti-cancer Activity of Centipede minima Extract in Triple-Negative Breast Cancer via Inhibition of AKT, NF-kappaB, and STAT3 Signaling Pathways.
Lee MM, Chan BD, Wong WY, Qu Z, Chan MS, Leung TW, Lin Y, Mok DK, Chen S, Tai WC.
Front Oncol. 2020 Apr 9; 10: 491. doi: 10.3389 / fonc.2020.00491
Inhibitory ASIC2-mediated calcineurin / NFAT against colorectal cancer by triterpenoids extracted from Rhus Chinensis Mill.
Wang G, Wang YZ, Yu Y, Wang JJ.
J Ethnopharmacol. 2019 May 10; 235: 255-267. doi: 10.1016 / j.jep.2019.02.029
Inhibition of metastasis and angiogenesis in Hep-2 cells by wheatgrass extract - an in vitro and in silico approach.
Shakya G, Balasubramanian S, Hoda M, Rajagopalan R.
Toxicol Mech Methods. 2018 Mar; 28 (3): 205-218. doi: 10.1080 / 15376516.2017.1388460
Anticancer Activity of Modified Tongyou Decoction on Eca109 Esophageal Cancer Cell Invasion and Metastasis through Regulation of the Epithelial-Mesenchymal Transition Mediated by the HIF-1alpha-Snail Axis.
Jia Y, Yan X, Cao Y, Song W, Zhang G, Hu X.
Evid Based Complement Alternat Med. 2020 Sep 29; 2020: 3053506. DOI: 10.1155 / 2020/3053506
The adjuvant value of Andrographis paniculata in metastatic esophageal cancer treatment - from preclinical perspectives.
Li L, Yue GG, Lee JK, Wong EC, Fung KP, Yu J, Lau CB, Chiu PW.
Sci Rep. 2017 Apr 12; 7 (1): 854. doi: 10.1038 / s41598-017-00934-x
The novel herbal cocktail MA128 suppresses tumor growth and the metastatic potential of highly malignant tumor cells.
Kim A, Im M, Yim NH, Hwang YH, Yang HJ, Ma JY.
Oncol Rep. 2015 Aug; 34 (2): 900-12. doi: 10.3892 / or.2015.4018.
The last problem that is still observed in the reviewed manuscript is the approach to the topic. I understand that the authors decided to carry out this paper by dividing the work into individual cancers. Apart from that idea that does not allow for a comparative analysis of the effects of compounds on similar pathways in various cancers, unfortunately, the presented data is often very superficial, and the authors only quote the facts without any analysis of the presented data and not attempt to draw more in-depth analyzes. In my opinion, the work should be rewritten and presented in a more careful way that allows the reader to find meaningful analyzes in it.
I encourage the authors to do that work do so due to the importance of the presented subject.

Author Response

Thank you for your comments to improve the quality of this review.

-Despite the clarification of the manuscript title, it does not precisely indicate the contained issues. Moreover, it suggests that the authors carried out a much more in-depth analysis than is the case. The title should still be corrected.

(Response): Thanks. We changed the title as your suggestion.

-Secondly, the narrowing of the topic is entirely incomprehensible. Why did the authors choose only works from the last five years?

(Response): It would be nice if we could include more expansive years of studies, however, reviewing the past five years of research is a period setting intensively keeping up-to-date with the latest findings. Therefore, it has been also adopted by many other studies as some examples listed below.

1) Are there advances in pharmacotherapy for panic disorder? A systematic review of the past five years. Daniela Caldirola, Alessandra Alciati, Alice Riva, Giampaolo Perna. 2018 Aug;19(12):1357-1368. doi: 10.1080/14656566.2018.1504921

2) Epidemiology of depression and diabetes: a systematic review. Tapash Roy, Cathy E Lloyd. 2012 Oct;142 Suppl:S8-21. doi: 10.1016/S0165-0327(12)70004-6.

3) Allosteric inhibitors of SHP2: an updated patent review (2015-2020). Jingwei Wu , Huan Zhang, Guilong Zhao, Runling Wang. 2020 Sep 28. doi: 10.2174/1568011817666200928114851. 

4) Significance of nanomaterials in electrochemical glucose sensors: An updated review (2016-2020). Ekin Sehit, Zeynep Altintas. 2020 Jul 1;159:112165. doi: 10.1016/j.bios.2020.112165.

5) Reirradiation of brain metastasis: Review of the last five years. M-N Nguyen, G Noel, D Antoni. 2019 Oct;23(6-7):531-540. doi: 10.1016/j.canrad.2019.07.144.

6) Anticancer activities of TCM and their active components against tumor metastasis, Wang K, Chen Q, Shao Y, Yin S, Liu C, Liu Y, Wang R, Wang T, Qiu Y, Yu H. Biomed Pharmacother. 2021 Jan;133:111044. doi: 10.1016/j.biopha.2020.111044.

As can be seen in this list of papers, there had been a number of studies reviewing the past five years, the focusing on topic of five years is quite reasonable.

-After all, they omitted some significant research and unconsciously mislead the reader, who, starting their interest in this important subject, may get the wrong impression that only these presented compounds, mixtures, extract were studied in the context of cancer metastasis.

(Response): Thank you for your suggestion. As you pointed out, we double-checked the omitted research you suggested again. We described the reasons of exclusion in each study below.

-After all, review papers are not once an impulse to new research directions. Such limitations are wrong. However, if the authors still agree to their concept, they should refer to other, more broadly discussing the subject of review papers, encouraging the reader to refer to them. This aspect is very much missing here.

(Response): This review is not once an impulse to new research direction. As described above, there are several review articles using the methods we used. However, we added and revised the manuscript as your comments in line 886-923.

- Besides, even recognizing the authors' right to narrow down the subject, there is a lack of at least a few of the manuscripts that appear in Pubmed and meet the given criteria. Even if

-Anti-cancer Activity of Centipede minima Extract in Triple-Negative Breast Cancer via Inhibition of AKT, NF-kappaB, and STAT3 Signaling Pathways.
Lee MM, Chan BD, Wong WY, Qu Z, Chan MS, Leung TW, Lin Y, Mok DK, Chen S, Tai WC.
Front Oncol. 2020 Apr 9; 10: 491. doi: 10.3389 / fonc.2020.00491

(Response): It already had been cited in our paper. You can find that in reference no. [118].

-Inhibitory ASIC2-mediated calcineurin / NFAT against colorectal cancer by triterpenoids extracted from Rhus Chinensis Mill.
Wang G, Wang YZ, Yu Y, Wang JJ.
J Ethnopharmacol. 2019 May 10; 235: 255-267. doi: 10.1016 / j.jep.2019.02.029
(Response): This paper dealt with Triterpenoids, which are compounds. However, we excluded “compounds” research as we described in the Methods part. Thus, this paper is not appropriate with our criteria.

-Inhibition of metastasis and angiogenesis in Hep-2 cells by wheatgrass extract - an in vitro and in silico approach.
Shakya G, Balasubramanian S, Hoda M, Rajagopalan R.
Toxicol Mech Methods. 2018 Mar; 28 (3): 205-218. doi: 10.1080 / 15376516.2017.1388460

(Response): This paper is related with Laryngeal cancer, which is not included in the five major cancers; lung, colorectal, gastric, liver and breast cancer.

-Anticancer Activity of Modified Tongyou Decoction on Eca109 Esophageal Cancer Cell Invasion and Metastasis through Regulation of the Epithelial-Mesenchymal Transition Mediated by the HIF-1alpha-Snail Axis.
Jia Y, Yan X, Cao Y, Song W, Zhang G, Hu X.
Evid Based Complement Alternat Med. 2020 Sep 29; 2020: 3053506. DOI: 10.1155 / 2020/3053506, The adjuvant value of Andrographis paniculata in metastatic esophageal cancer treatment - from preclinical perspectives.
Li L, Yue GG, Lee JK, Wong EC, Fung KP, Yu J, Lau CB, Chiu PW.
Sci Rep. 2017 Apr 12; 7 (1): 854. doi: 10.1038 / s41598-017-00934-x

(Response): We also looked at the above two papers and we could find that these papers were related with Esophageal cancer, which is not included in the five major cancers; lung, colorectal, gastric, liver and breast cancer.

The novel herbal cocktail MA128 suppresses tumor growth and the metastatic potential of highly malignant tumor cells.
Kim A, Im M, Yim NH, Hwang YH, Yang HJ, Ma JY.
Oncol Rep. 2015 Aug; 34 (2): 900-12. doi: 10.3892 / or.2015.4018.

(Response): It already had been cited in our paper. You can find that in reference no. [59].

-The last problem that is still observed in the reviewed manuscript is the approach to the topic. I understand that the authors decided to carry out this paper by dividing the work into individual cancers. Apart from that idea that does not allow for a comparative analysis of the effects of compounds on similar pathways in various cancers, unfortunately, the presented data is often very superficial, and the authors only quote the facts without any analysis of the presented data and not attempt to draw more in-depth analyzes. In my opinion, the work should be rewritten and presented in a more careful way that allows the reader to find meaningful analyzes in it. I encourage the authors to do that work do so due to the importance of the presented subject.

(Response): We analyzed the articles by their drugs, experimental model, dose/duration, efficacy, mechanism in table and the specific mechanisms were described in results section. Can you point the part which is superficial? Also, a total of 79 of anti-metastatic effects of herbal medicines were reviewed in this study. In fact, two other reviewers pointed that this manuscript is quite long and one of them suggested to report the results of the most interesting papers first and to summarize other data shortly later. However, we chose not to, because that modification could make it superficial and subjective while choosing the interesting papers which could lead readers to wrong direction.

Unlike other diseases, cancer is characterized by metastasis to various part of body. While there had been many specific studies of certain cancers, comprehensive studies dealing with multiple cancer species have not been established for metastatic cancer. The purpose of this review is to analyze the existing data of the five recent years about the efficacy, mechanisms and new strategies of herbal medicine against metastasis in five major cancer.

Also, our research has its own meaning.

  1. First of all, by analyzing five major cancer studies in recent five years, we could review the major mechanisms and some research trends of anti-metastatic effects of herbal medicine, which were EMT, MMPs, ROS-related process and signaling pathway. We described each frequent mechanism in Discussion part and revealed it through a schematic diagram in figure 7.
  2. The individual mechanisms of each paper are presented as diagrams in integrated view. Since the action of herbal medicine is not determined by one mechanism, integrating multiple mechanisms can help the readers to understand the roles of herbal medicines in metastasis.
  3. Moreover, herbal medicines that can be used for metastasis in cancer were suggested. The five major cancers have a high prevalence as well as a high mortality rate caused by metastasis. In particular, there have been many cases of metastasis between the five major cancers, hence it is meaningful enough to present an herbal medicine that can be used in such cases like figure 6.
  4. Also, we found that there are only a few articles on clinical studies, which is only eight in entire articles. Thus, our study fully emphasized the requirement of more experiments for clinical utilization of herbal medicine in further studies and supported the data base for researchers who eager to develop novel drugs for metastasis.

Round 3

Reviewer 1 Report

The clarification of the manuscript title meant that the paper is in line with the presented subject. The authors also introduced corrections which cause me to accept the work in its current form.